# DTP: Delta-Guided Two Stage Pruning for Mamba-based Multimodal Large Language Models

**Seong Yeol Park, Min Jung Kwon, Xianghua Piao, Yeong Hyeon Gu**[*]
Sejong University
{seongyeol, mjkwon, hianghua}@sju.ac.kr
yhgu@sejong.ac.kr

## Abstract

Multimodal large language models built on the Mamba architecture offer efficiency advantages, yet remain hampered by redundant visual tokens that inflate inference cost, with the prefill stage accounting for the majority of total inference time. We introduce Delta-guided Two stage Pruning (DTP), a method that progressively reduces token redundancy through selective pruning at early layer and complete pruning at late layer. Unlike Transformer-oriented pruning methods, our approach derives token importance directly from Mamba's internal parameters. The statistical distribution of these importance scores, combined with implicit attention patterns, then provides the basis for determining both the pruning layers and the tokens to be removed. Extensive evaluation across diverse benchmarks shows that DTP cuts computation by nearly 50%, maintains higher task performance than existing pruning methods, and further achieves over a 35% reduction in prefill latency. Beyond efficiency, our analysis reveals previously underexplored behaviors of visual tokens within Mamba layers, suggesting a principled perspective for designing future pruning techniques in Mamba-based Multimodal Large Language Models.

## 1 Introduction

Multimodal Large Language Models (MLLMs) are capable of jointly understanding and generating across different modalities such as images and text, thereby handling complex tasks that are difficult for single-modality models (Liu et al., 2023; Dai et al., 2023; Peng et al., 2023; Wu et al., 2024). These capabilities have demonstrated outstanding performance in diverse tasks such as visual question answering (Guo et al., 2023; Hu et al., 2024; Fang et al., 2025; Wang et al., 2025; Dong et al., 2025) and reasoning segmentation (Lai et al., 2024; Ren et al., 2024; Xia et al., 2024).

Most existing MLLMs are built upon the Transformer architecture (Vaswani et al., 2017), which has shown strong performance in a wide range of multimodal tasks through the self-attention mechanism. However, self-attention requires computing interactions between all token pairs, leading to $O(n^2)$ time complexity that must be repeatedly incurred at every step when generating new tokens. To mitigate this redundant computation, the KV-Cache technique has been introduced, which effectively accelerates autoregressive generation during inference. Nevertheless, KV-Cache requires storing key-value pairs at every generation step, which results in substantial memory consumption. Furthermore, it does not reduce the computational cost of the prefill stage.

To overcome these structural limitations of Transformers, the recently proposed Mamba architecture (Gu & Dao, 2023) leverages State Space Models (SSMs) to recurrently update hidden states, thereby achieving linear-time complexity. Figure 1 contrasts the inference process between Transformer-based and Mamba-based MLLMs. As shown in Figure 1 (a), Transformers must recompute self-attention with all previous tokens whenever a new token is generated, causing decoding costs to increase linearly with sequence length. In contrast, as illustrated in Figure 1 (b), Mamba generates the next token through a single-step hidden state update without revisiting the entire input sequence.

---

[*]Corresponding author.

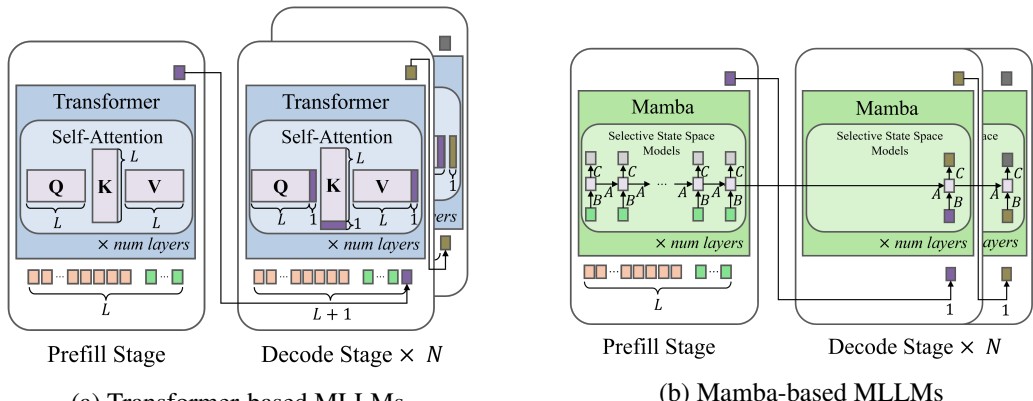

Figure 1: Comparison of inference structures between Transformer-based and Mamba-based MLLMs.

This structural difference allows Mamba-based MLLMs to achieve much lower memory usage and faster decoding compared to Transformer-based models, a benefit that has been empirically validated in recent studies (Liu et al., 2024; Qiao et al., 2024; Zhao et al., 2025).

Thus, while Mamba provides clear efficiency advantages in the decoding stage, the majority of inference time is still spent in the prefill stage, where all input tokens must be processed initially. The inefficiency of the prefill stage is particularly pronounced in multimodal settings, since the number of visual tokens far exceeds that of text tokens, greatly increasing the overall input length. However, many of these visual tokens are redundant or uninformative, and often do not contribute to the final output (Chen et al., 2024). Based on this observation, various pruning methods have been proposed in Transformer-based MLLMs to reduce computational cost by removing unnecessary visual tokens (Alvar et al., 2025; Yang et al., 2025; Ye et al., 2025a;b; Lin et al., 2025). While effective in improving efficiency, these approaches rely on attention scores to estimate token importance, making them inherently dependent on the Transformer architecture and not directly applicable to Mamba.

To address this limitation, we propose a novel visual token pruning framework tailored for Mamba-based MLLMs. The proposed DTP (Delta-guided Two stage Pruning) leverages the input-dependent parameter $\Delta_t$ to estimate token importance and performs pruning during inference without any re-training. Pruning is applied at two specific layers, with their positions determined from the observed distribution of token importance and the analysis of Mamba's implicit attention matrices. Based on this rationale, our design effectively removes unnecessary tokens while incurring smaller performance degradation compared to other approaches.

Through extensive experiments on representative Mamba-based MLLMs, Cobra (Zhao et al., 2025) and RoboMamba (Liu et al., 2024), we demonstrate that DTP reduces computational cost by nearly half while maintaining comparatively high performance relative to existing methods. Furthermore, we identify the optimal internal parameters in Mamba for evaluating token importance, thereby maximizing the effectiveness of pruning. In addition to empirical results, our study provides new insights into the role and significance of visual tokens across layers in Mamba-based MLLMs, offering an effective methodology for visual token pruning.

Our main contributions are summarized as follows:

- We propose DTP (Delta-guided Two stage Pruning), the novel visual token pruning framework designed for Mamba-based MLLMs. DTP leverages the input-dependent parameter $\Delta_t$ to estimate token importance and performs visual token pruning during inference without requiring additional training.

- Pruning is applied at two specific layers, with their positions determined based on token importance distribution and the analysis of Mamba's implicit attention matrices. This design enables the effective removal of redundant visual tokens while maintaining stable performance.

- Extensive experiments on Cobra and RoboMamba demonstrate that DTP reduces FLOPs by nearly half with less performance degradation compared to existing methods. In addition, we identify the optimal internal parameters for token importance estimation, providing further insights into the role of visual tokens across layers in Mamba-based MLLMs.

## 2 RELATED WORK

**Token Reduction in Mamba-based Model**. Token reduction has been studied in Transformer-based models as a way to alleviate high computational cost and accelerate inference by removing unnecessary input tokens without causing significant performance degradation (Bolya et al., 2022; Kong et al., 2022; Haurum et al., 2023; Wei et al., 2023; Kim et al., 2024; 2025). However, since Mamba is based on State Space Models (SSMs) rather than the self-attention mechanism central to Transformers, token reduction techniques designed for Transformer architectures cannot be directly applied. Accordingly, recent studies have proposed token reduction methods specifically tailored to the structural characteristics of Mamba. Zhan et al. (2024) proposed a token pruning method for Vision Mamba (ViM) (Zhu et al., 2024) and PlainMamba (Yang et al., 2024), introducing a pruning-aware hidden state alignment approach to stabilize the neighborhoods of the remaining tokens and a token importance estimation mechanism specific to Mamba, thereby improving inference speed while minimizing performance degradation. Shen et al. (2024) proposed Famba-V, a cross-layer token fusion method for ViM that identifies and merges similar tokens across layers, improving training efficiency while maintaining a balance with accuracy. Although these works explore diverse token reduction strategies such as pruning and fusion in Mamba-based models, they remain limited to unimodal vision tasks, and token reduction in Mamba-based MLLMs has not yet been sufficiently explored.

**Mamba-based MLLMs**. Recent studies have sought to leverage the structural efficiency of Mamba by extending Transformer-based MLLMs into Mamba-based MLLMs, aiming to achieve faster inference speed and improved efficiency in long-sequence processing. Qiao et al. (2024) proposed VL-Mamba, the first multimodal architecture that replaces the Transformer-based language model with a Mamba language model. It adopts SigLIP (Zhai et al., 2023) as the vision encoder and introduces the Vision Selective Scan (VSS) module within a multimodal connector to enhance representational capacity, demonstrating competitive results across a variety of multimodal benchmarks. Liu et al. (2024) introduced RoboMamba, which combines a CLIP vision encoder (Radford et al., 2021) with Mamba and adds it with a lightweight policy head to enable SE(3) pose prediction and vision-language-action modeling. In addition to its strong performance in robotic manipulation tasks, it also exhibits remarkable multimodal reasoning capability. Furthermore, Zhao et al. (2025) presented Cobra, which integrates a pre-trained Mamba language model with visual encoders such as DINOv2 (Oquab et al., 2023) and SigLIP (Zhai et al., 2023). Compared to Transformer-based MLLMs, it achieves both faster inference speed and superior performance.

## 3 PRELIMINARIES

### 3.1 STATE SPACE MODELS AND MAMBA

State Space Models (SSMs) are the core structure of Mamba (Gu & Dao, 2023), which transform an input sequence $x(t) \in \mathbb{R}$ into an output sequence $y(t) \in \mathbb{R}$ through a hidden state $h(t) \in \mathbb{R}^N$, and are defined as:

$$h'(t) = Ah(t) + Bx(t), \quad y(t) = Ch(t), \tag{1}$$

where $A$ governs the state transitions, $B$ maps the input into the hidden state, and $C$ projects the hidden state into the output sequence.

In this basic form, SSMs have the limitation of linear time invariance (LTI), in which the same fixed parameters are applied to every time step of the input sequence, making it impossible to selectively process important or redundant information. In addition, while this formulation is designed for continuous systems, deep learning models generally operate on discrete systems, and therefore the continuous parameters must be discretized.

To address this issue, Mamba introduces an input-dependent parameter $\Delta_t$. This parameter is computed from the input at each time step through a linear transformation followed by the softplus

function, which enables different state updates at each step. By leveraging this input-dependent parameter, Mamba discretizes SSMs and proposes Selective SSMs (S6), which allow inputs to be processed selectively at each time step, as formulated below:

$$\bar{A}_t = \exp(\Delta_t A), \quad \bar{B}_t = (\Delta_t A)^{-1} \left(\exp(\Delta_t A) - I\right) \Delta_t B$$
$$h_t = \bar{A}_t h_{t-1} + \bar{B}_t x_t, \quad y_t = C_t h_t \tag{2}$$

### 3.2 ATTENTION MATRICES IN MAMBA

Ali et al. (2024) demonstrated that the selective SSM layer can be unfolded into a causal kernel that closely resembles the attention matrix in Transformers. This finding provides an interpretation of Mamba's selective SSMs as implicitly incorporating attention-like behavior, even without an explicit attention mechanism. Specifically, the selective SSM formulation can be expanded into a convolutional form, yielding a kernel that functions as implicit attention weights. From Equation 2, assuming the initial state $h_0 = 0$, the output $y_t$ can be written as:

$$y_t = \sum_{j=1}^{t} C_t \left(\prod_{k=j+1}^{t} \bar{A}_k\right) \bar{B}_j x_j \tag{3}$$

By defining

$$K_{t,j} = C_t \left(\prod_{k=j+1}^{t} \bar{A}_k\right) \bar{B}_j, \quad y_t = \sum_{j=1}^{t} K_{t,j} x_j, \tag{4}$$

we see that $K_{t,j}$ represents the coefficient through which the input $x_j$ is linearly transformed and incorporated into the output at time step $t$.

By arranging all coefficients, we obtain the following lower-triangular implicit attention matrix:

$$K = \begin{bmatrix} C_1\bar{B}_1 & 0 & 0 & \cdots & 0 \\ C_2\bar{A}_2\bar{B}_1 & C_2\bar{B}_2 & 0 & \cdots & 0 \\ C_3\bar{A}_2\bar{A}_3\bar{B}_1 & C_3\bar{A}_3\bar{B}_2 & C_3\bar{B}_3 & \cdots & 0 \\ \vdots & \vdots & \vdots & \ddots & \vdots \\ C_L\left(\prod_{k=2}^{L}\bar{A}_k\right)\bar{B}_1 & C_L\left(\prod_{k=3}^{L}\bar{A}_k\right)\bar{B}_2 & C_L\left(\prod_{k=4}^{L}\bar{A}_k\right)\bar{B}_3 & \cdots & C_L\bar{B}_L \end{bmatrix} \tag{5}$$

This matrix $K \in \mathbb{R}^{L \times L}$, where $L$ denotes the sequence length, can be interpreted in a way similar to the attention matrix of Transformers, where each row corresponds to the output at time $t$ and each column shows how an input $x_j$ propagates its influence to subsequent outputs.

## 4 METHOD

In this section, we propose DTP (Delta-guided Two stage Pruning), a pruning strategy for Mamba-based MLLMs. As shown in Figure 2, DTP follows a two stage pruning strategy, performing selective pruning in the early layer and complete pruning in the late layer. To determine the specific layers where pruning is applied and to conduct selective pruning in the early layer, we leverage $\Delta_t$, the key parameter that enables the selectivity of Mamba, to evaluate the importance of visual tokens in each Mamba block.

### 4.1 TOKEN IMPORTANCE FROM $\Delta_t$

As described in Section 3.1, LTI SSMs apply the same parameters to all input sequences and therefore lack the selectivity to distinguish the relative importance of tokens. Accordingly, Mamba (Gu & Dao, 2023) introduces an input-dependent parameter $\Delta_t$ derived from the input sequence, and $\Delta_t$ serves as the key mechanism that enables selectivity by discretizing continuous SSMs and controlling the state transition matrix $\bar{A}_t$ and the input mapping matrix $\bar{B}_t$. Specifically, $\Delta_t$ directly modulates both $\bar{A}_t$ and $\bar{B}_t$, thereby controlling the discretized state transition dynamics and determining the extent to which each token contributes to subsequent hidden states. Tokens with larger

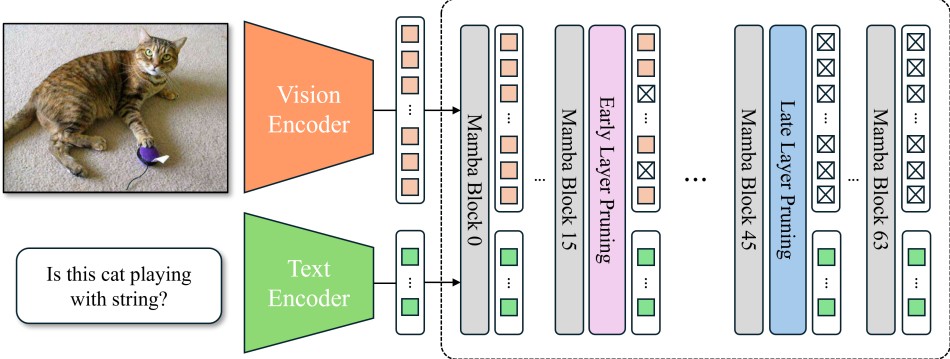

Figure 2: Overview of the proposed DTP(Delta-guided Two stage Pruning) method.

$\Delta_t$ values induce stronger state updates, whereas those with smaller values yield only minor transitions, effectively acting as less influential inputs. Building on this role, we directly leverage $\Delta_t$ to obtain token importance scores for visual token pruning in Mamba-based MLLMs. The importance score of the $j$-th token, denoted as $s_j$, is defined as follows:

$$s_j = \frac{1}{D}\sum_{d=1}^{D}\Delta_{j,d},\qquad(6)$$

where $D$ is the dimension size. This allows token importance to be directly estimated from $\Delta_t$ during inference, without requiring any additional training or modification. Based on the token importance scores $s$ derived from $\Delta_t$, we adopt a top-$k$ selection strategy to retain the most informative visual tokens. At the pruning layers, visual tokens are ranked according to their importance scores, and only the top $k$ tokens are preserved while the others are discarded. This reduces redundant visual information and enables computation to focus on the more critical tokens during reasoning. For a more comprehensive analysis, we further compared the proposed token importance parameter $\Delta_t$ with other internal parameters such as $y_t$, $B_t$, and $C_t$, and the results confirmed that $\Delta_t$ is the most suitable criterion for evaluating the importance of tokens in pruning.

## 4.2 Pruning Strategy

Drawing on prior studies of token reduction in Transformer-based MLLMs (Chen et al., 2024; Lin et al., 2025; Ye et al., 2025a), which observed that visual tokens contain redundant information and exhibit minimal attention in deeper layers, we apply pruning specifically to visual tokens. Token pruning improves computational efficiency and latency by removing redundant tokens, and in deeply stacked MLLM architectures, the process of identifying the optimal pruning point is just as important as determining the parameter used to measure token importance.

Based on the token-importance parameter $\Delta_t$ introduced in Section 4.1, we conduct a preliminary analysis to determine which layers are suitable candidates for pruning. To quantify how the importance of tokens varies within each layer, we compute the layer-wise standard deviation of $\Delta_t$-guided token importance, defined as:

$$Std_\ell = \sqrt{\frac{1}{N}\sum_{j=1}^{N}\left(s_{j,\ell} - \bar{s}_\ell\right)^2}\qquad(7)$$

where $s_{j,\ell}$ denotes the importance value of the $j$-th token at layer $\ell$, $\bar{s}_\ell$ is the mean importance at that layer, and $N$ is the number of tokens.

For this preliminary analysis, we randomly sample 50 examples from each benchmark dataset used in our performance evaluation and compute the standard deviation for all 64 layers in Cobra and RoboMamba. The full results are presented in Appendix B. Across all datasets, both models exhibit highly consistent trends, demonstrating that layer-wise standard deviation provides an efficient and reliable criterion for identifying promising pruning layers without the need for additional heuristics or layer-specific thresholds.

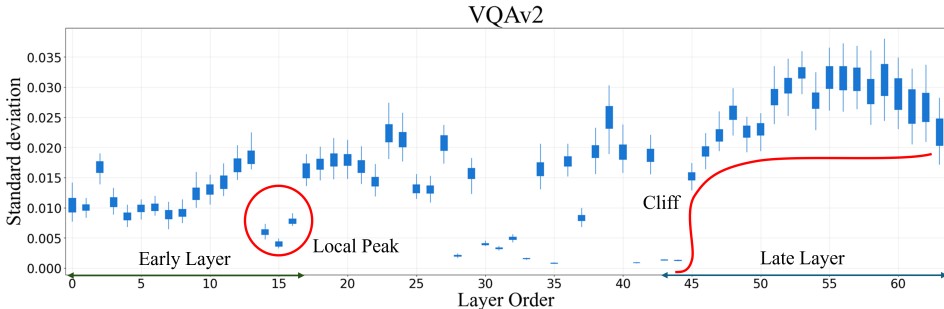

Figure 3: Layer-wise standard deviation of token-importance values for the Mamba-based MLLM Cobra (Zhao et al., 2025) on VQAv2. The early layers contain a clear local peak, while the late layers exhibit a cliff

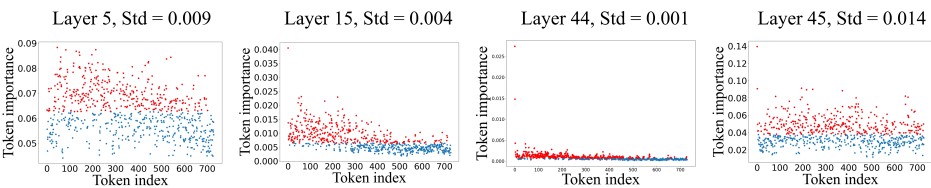

Figure 4: Visualization of token importance distributions at different layers of a Mamba-based MLLM, Cobra (Zhao et al., 2025). Red and blue points denote tokens within and outside the top 50% by importance, respectively.

In addition to the standard deviation analysis, we further examine the distribution of token-importance values derived from the parameter described in Section 4.1. Moreover, inspired by the implicit attention pattern of Mamba layers discussed in Section 3.2, we incorporate an additional analysis to intuitively capture token–token interaction behaviors.

**Selective pruning at the early layer.** Pruning at very early layers has the advantage of achieving high computational efficiency, but it carries a significant risk of discarding tokens that could later serve as meaningful information in deeper layers. In addition, at such early layers, the distinction of token importance is not yet clear, making reliable selection difficult. To mitigate these issues, we focus on the first 25% of layers, guided by the preliminary analysis presented earlier, and examine how token importance patterns evolve within this region.

As shown in Figure 3, the standard deviation of token importance exhibits a noticeable local peak in this early interval, where the standard deviation is relatively low compared to neighboring layers. Figure 4 further illustrates that layers near this peak contain a cluster of tokens with similar importance values, along with a small subset of tokens that hold comparatively higher importance. When examining the implicit attention pattern in Figure 5, we observe that in layer 5, which shows a relatively high standard deviation before the local peak appears, there is no clear token–token interaction pattern in the lower-triangular region. In contrast, in layer 15 within the local-peak region where the standard deviation becomes low, we can see that a token–token interaction pattern begins to form in the lower-triangular area. Considering all the numerical and quantitative analyses that demonstrate consistent patterns, we find that layers with low standard deviation values exhibit implicit attention patterns with strong token-token interaction and represent points where important tokens can be separated.

Leveraging these observations, we select the optimal early pruning layer $k_{\text{early}}$ as the layer within the first 25% of the model depth in a model with $L$ layers that exhibits the minimum standard deviation of token importance values:

$$k_{\text{early}} = \operatorname*{argmin}_{\ell} Std_\ell, \qquad \text{where } 0 \le \ell \le 0.25L. \tag{8}$$

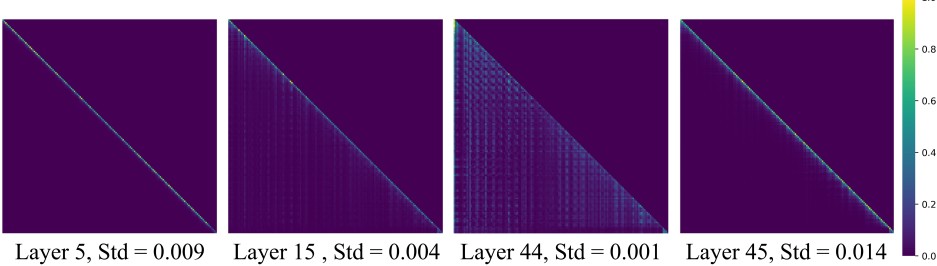

Figure 5: Visualization of implicit attention patterns across different layers in Mamba-based MLLM, Cobra (Zhao et al., 2025). Appendix C provides an overview of implicit attention patterns across all layers.

**Complete Pruning at Late Layers**. As shown in Figure 3, the standard deviation of token importance values increases sharply beyond roughly 70% of the model depth, producing a cliff-like pattern that marks the onset of a high standard deviation regime. This observation is further supported by the token importance distributions in Figure 4, which become increasingly diffuse and irregular in late layers, lacking the coherent clustering observed earlier. When examining the implicit attention pattern in Figure 5, we observe that at the cliff point where the difference in standard deviation between adjacent layers is the largest, layer 44, which exhibits a very low standard deviation, shows a clear token–token interaction pattern in the lower-triangular region. In contrast, the next layer, layer 45, displays a sharp increase in standard deviation, and the token–token interaction pattern in the lower-triangular area begins to weaken. Taken together, these signals indicate that once the model enters this high standard deviation regime, no longer provide meaningful cues for selecting salient tokens. In other words, the implicit attention structure collapses to the point that partial pruning becomes unreliable. Consequently, complete pruning is more effective than selective pruning in this stage of the model, as visual tokens no longer contribute stable or discriminative information to the downstream computation.

Leveraging these observations, we select the optimal late pruning layer $k_{\text{late}}$ as the layer within the final 30% of the model depth in a model with $L$ layers that exhibits the maximum inter-layer change in the standard deviation of token importance values:

$$k_{\text{late}} = \underset{\ell}{\operatorname{argmax}} \, | \, Std_{\ell+1} - Std_{\ell} \, | + 1, \qquad \text{where } 0.7L \leq \ell < L - 1. \tag{9}$$

## 5 EXPERIMENT

### 5.1 EXPERIMENTAL SETUP

We evaluate our pruning strategy on two representative Mamba-based MLLMs: Cobra (Zhao et al., 2025) and RoboMamba (Liu et al., 2024). These models are representative Mamba-based MLLMs and are used as baselines since their pretrained weights are publicly available and they provide strong performance across multimodal tasks. For Cobra, we apply our method to six benchmarks that cover diverse aspects of multimodal reasoning, including GQA (Hudson & Manning, 2019), VQAv2 (Goyal et al., 2017), TextVQA (Singh et al., 2019), POPE (Li et al., 2023), VSR (Liu et al., 2023), and VizWiz (Gurari et al., 2018). For RoboMamba, we evaluate on five benchmarks, including OKVQA (Marino et al., 2019), GQA (Hudson & Manning, 2019), VQAv2 (Goyal et al., 2017), POPE (Li et al., 2023), and VizWiz (Gurari et al., 2018).

The baselines considered in this study are three representative token pruning methods originally proposed for Transformer-based MLLMs: (1) FastV (Chen et al., 2024) prunes visual tokens at a designated layer $k$ based on attention scores, with the pruning ratio controlled by $r$. Since Mamba-based MLLMs do not expose explicit attention scores, we replace them with our proposed $\Delta_t$-based token importance measure for benchmarking. (2) VTW (Lin et al., 2025) determines the optimal layer $k$ for withdrawing visual tokens by sampling a small subset of the dataset, comparing the original output with the output after token withdrawal, and selecting the earliest layer where the KL

divergence between the two logits falls below a predefined threshold. As this method provides a general criterion that is independent of model architecture, it also serves as an appropriate baseline for our study. (3) DART (Wen et al., 2025) defines a small subset of visual tokens as pivot tokens and removes the remaining tokens based on their similarity to these pivots, with the reduction controlled by the keep ratio $r$. All experiments are implemented in PyTorch and executed on a single NVIDIA RTX 5090 GPU to ensure fair comparisons.

Table 1: Comparison of pruning methods on the Cobra model. The table presents FLOPs, FLOPs ratio, evaluation scores across six benchmarks, and the average score.

| Method | FLOPs | FLOPs ratio | GQA | VQAv2 | TextVQA | POPE | VSR | VizWiz | Avg |
|---|---|---|---|---|---|---|---|---|---|
| Baseline (Cobra) | 2.01 | 100% | 62.3 | 77.8 | 58.2 | 88.4 | 58.4 | 49.7 | 65.8 |
| FastV ($k$=2, $r$=0.7) | 1.45 | 72% | 62.1 (−0.2) | 77.4 (−0.4) | 56.9 (−1.3) | 87.7 (−0.7) | 58.0 (−0.4) | 49.8 (+0.1) | 65.3 (−0.5) |
| VTW ($k$=45) | 1.43 | 71% | 62.1 (−0.2) | 77.7 (−0.1) | 58.2 (+0.0) | 88.3 (−0.1) | 58.5 (+0.1) | 49.5 (−0.2) | 65.7 (−0.1) |
| DART ($r$=0.66) | 1.38 | 69% | 62.0 (−0.3) | 77.1 (−0.7) | 57.0 (−1.2) | 87.4 (−1.0) | 58.2 (−0.2) | 49.7 (+0.0) | 65.2 (−0.6) |
| Ours ($r$=0.9) | 1.35 | 67% | 62.0 (−0.3) | 77.7 (−0.1) | 57.9 (−0.3) | 88.3 (−0.1) | 58.9 (+0.5) | 49.7 (+0.0) | 65.8 (+0.0) |
| FastV ($k$=2, $r$=0.5) | 1.06 | 53% | 61.7 (−0.6) | 76.8 (−1.0) | 55.0 (−3.2) | 87.4 (−1.0) | 57.3 (−1.1) | 50.1 (+0.4) | 64.7 (−1.1) |
| VTW ($k$=32) | 1.04 | 52% | 47.1 (−15.2) | 54.1 (−23.7) | 42.6 (−15.6) | 74.1 (−14.3) | 57.9 (−0.5) | 48.5 (−1.2) | 54.0 (−11.8) |
| DART ($r$=0.44) | 0.96 | 48% | 61.2 (−1.1) | 76.1 (−1.7) | 55.1 (−3.1) | 86.3 (−2.1) | 57.7 (−0.7) | 49.5 (−0.2) | 64.3 (−1.5) |
| Ours ($r$=0.5) | 0.97 | 48% | 61.4 (−0.9) | 77.1 (−0.7) | 56.1 (−2.1) | 87.3 (−1.1) | 57.9 (−0.5) | 49.6 (−0.1) | 64.9 (−0.9) |

Table 2: Comparison of pruning methods on the RoboMamba model. The table presents FLOPs, FLOPs ratio, evaluation scores across five benchmarks, and the average score.

| Method | FLOPs | FLOPs ratio | OKVQA | GQA | VQAv2 | POPE | VizWiz | Avg |
|---|---|---|---|---|---|---|---|---|
| Baseline (RoboMamba) | 0.70 | 100% | 64.4 | 56.4 | 74.9 | 85.2 | 54.2 | 67.0 |
| FastV ($k$=2, $r$=0.7) | 0.50 | 71% | 64.1 (−0.3) | 56.1 (−0.3) | 74.4 (−0.5) | 85.3 (+0.1) | 53.0 (−1.2) | 66.6 (−0.4) |
| VTW ($k$=45) | 0.50 | 71% | 64.0 (−0.4) | 55.9 (−0.5) | 74.7 (−0.2) | 85.3 (+0.1) | 53.3 (−0.9) | 66.6 (−0.4) |
| DART ($r$=0.63) | 0.47 | 67% | 64.2 (−0.2) | 55.4 (−1.0) | 73.9 (−1.0) | 84.2 (−1.0) | 53.0 (−1.2) | 66.1 (−0.9) |
| Ours ($r$=0.9) | 0.46 | 66% | 64.2 (−0.2) | 56.1 (−0.3) | 74.7 (−0.2) | 85.2 (0.0) | 53.2 (−1.0) | 66.7 (−0.3) |
| FastV ($k$=2, $r$=0.5) | 0.37 | 53% | 63.4 (−1.0) | 55.1 (−1.3) | 73.4 (−1.5) | 84.2 (−1.0) | 52.1 (−2.1) | 65.6 (−1.4) |
| VTW ($k$=32) | 0.36 | 51% | 40.0 (−24.4) | 44.7 (−11.7) | 55.2 (−19.7) | 82.1 (−3.1) | 45.2 (−9.0) | 53.4 (−13.6) |
| DART ($r$=0.47) | 0.36 | 51% | 64.1 (−0.3) | 54.0 (−2.4) | 72.8 (−2.1) | 83.4 (−1.8) | 52.6 (−1.6) | 65.4 (−1.6) |
| Ours ($r$=0.5) | 0.34 | 49% | 63.8 (−0.6) | 54.9 (−1.5) | 73.6 (−1.3) | 84.4 (−0.8) | 52.8 (−1.4) | 65.9 (−1.1) |

## 5.2 MAIN RESULTS

We evaluate all pruning methods under two comparable FLOPs reduction regimes, a reduction of about 70% relative to the baseline and a reduction of about 50%. Table 1 presents the results on the Cobra model. For the 70% FLOPs level, FastV is configured with $k = 2$ and $r = 0.7$, achieving a FLOPs ratio of 72%. This configuration yields noticeable accuracy drops, including a reduction of 1.3 on TextVQA and a reduction of 0.7 on POPE. VTW is applied with its KL divergence based withdrawal criterion, which selects $k = 45$ and achieves a similar FLOPs ratio of 71% with only a 0.1 decrease in average accuracy. This indicates that the optimal $k$ determined by the withdrawal criterion of VTW indirectly supports the validity of our complete pruning at the late layer strategy described in Section 4.2. DART follows its original design, where it prunes at $k = 2$, retains eight

pivot tokens, and uses a keep ratio of $r = 0.66$. Under this setting, which results in approximately 69% FLOPs, DART exhibits clear degradation across several benchmarks, including a reduction of 1.2 on TextVQA, a reduction of 1.0 on POPE, and a reduction of 0.7 on VQAv2. Our method determines the pruning points using the preliminary analysis in Section 4.2, which consistently identified the early and late pruning layers as $k_{\text{early}} = 15$ and $k_{\text{late}} = 45$ across all benchmark datasets. With $r = 0.9$ applied at the early layer, DTP attains a FLOPs ratio of 67% while exactly matching the baseline's average score of 65.8, demonstrating substantially lower accuracy degradation than existing methods. When FLOPs were reduced to about half of the baseline, FastV is configured with $k = 2$ and $r = 0.5$ for comparison with existing methods. VTW is a method for finding the optimal k, we fixed k = 32 to enable comparison under this specific FLOPs condition. In this setting, FastV showed significant performance degradation, including a 3.2 point drop on TextVQA and a 1.1 point drop on VSR, while VTW exhibited large performance losses across all datasets. DART prunes at $k = 2$ with a keep ratio of $r = 0.47$, retaining eight pivot tokens. Under this configuration, which corresponds to roughly 48% FLOPs, DART also suffers noticeable accuracy degradation across benchmarks. Our method applies a keep ratio of $r = 0.5$ at the early pruning point $k_{\text{early}} = 15$, followed by complete pruning at $k_{\text{late}} = 45$. Our proposed method showed only a 0.9 point decrease on average compared to the baseline, demonstrating smaller performance degradation than the other methods.

Table 2 presents the results on RoboMamba. Our proposed method shows only a 0.3 point performance drop even when FLOPs are reduced to 66%. When FLOPs are further reduced to around 50%, VTW exhibits a severe performance degradation, similar to the case of Cobra. Both FastV, DART and our method experience some decrease in performance, but our method shows relatively smaller drops on OKVQA and VizWiz, achieving an average score of 65.9 with the least overall degradation. This effect arises because RoboMamba employs the CLIP vision encoder (Radford et al., 2021) with only 256 image tokens. Compared to Cobra, which processes 729 tokens, the smaller number of tokens makes RoboMamba more susceptible to information loss under aggressive pruning.

## 5.3 Efficiency Analysis

To verify that the FLOPs reduction achieved by DTP leads to practical runtime improvements, we additionally measure wall-clock inference latency on the Cobra model using the POPE benchmark. The evaluation includes mean prefill latency, mean decode latency, total inference time, and GPU memory consumption. As shown in Table 3, DTP reduces the mean prefill latency from 98.04 ms to 61.54 ms, corresponding to a 37% improvement, and lowers the total latency from 16m 05s to 10m 35s and GPU memory usage also decreases from 8.8 GB to 8.3 GB. Compared to existing pruning baselines such as FastV, VTW, and DART, DTP provides competitive or superior latency while maintaining strong accuracy and larger FLOPs reduction. These results confirm that the efficiency gains of DTP extend beyond computation savings and translate into substantial real-world speedups.

Table 3: Inference latency and gpu memory usage on the Cobra model using the POPE benchmark. The table reports mean prefill/decode latency, total inference time, and peak GPU memory consumption for different pruning methods.

| Method | FLOPs | Prefill Mean (ms) | Decode Mean (ms) | Total Latency | GPU Memory |
|---|---|---|---|---|---|
| Cobra | 2.01 | 98.04 | 5.04 | 16m 05s | 8.8 GB |
| FastV ($k=2$, $r=0.5$) | 1.06 | 60.64 | 4.94 | 10m 24s | 8.5 GB |
| VTW ($k=32$) | 1.04 | 67.31 | 4.99 | 11m 25s | 8.3 GB |
| DART ($r=0.44$) | 0.96 | 76.45 | 4.95 | 12m 44s | 8.5 GB |
| DTP ($r=0.5$) | 0.97 | 61.54 | 5.02 | 10m 35s | 8.3 GB |

## 5.4 Ablation Study

All ablation studies in this section are conducted under the setting where FLOPs are reduced to approximately 50% of the baseline.

**Identifying effective internal parameters for token importance.** Table 4 presents the ablation study results when different internal parameters of the selective SSM are used to compute token

Table 4: Ablation study on internal parameters of Mamba for token importance estimation.

| | Cobra | | | RoboMamba | | |
|---|---|---|---|---|---|---|
| Parameter | GQA | TextVQA | Vizwiz | GQA | POPE | OKVQA |
| $y_t$ | 61.1 | 48.0 | 49.0 | 54.5 | 83.8 | 63.8 |
| $B_t$ | 60.2 | 47.4 | 49.2 | 54.2 | 82.9 | 62.6 |
| $C_t$ | 58.9 | 44.9 | 49.6 | 53.0 | 81.6 | 62.0 |
| $\Delta_t$ | 61.4 | 56.1 | 49.6 | 54.9 | 84.4 | 63.8 |

Table 5: Ablation study on token selection methods for Cobra and RoboMamba

| | Cobra | | | RoboMamba | | |
|---|---|---|---|---|---|---|
| Method | GQA | TextVQA | Vizwiz | GQA | POPE | OKVQA |
| Random | 61.4 | 53.7 | 49.4 | 54.1 | 83.6 | 63.2 |
| Top-$k$ (all tokens) | 7.01 | 10.3 | 30.7 | 21.2 | 56.8 | 46.5 |
| Top-$k$ (visual only) | 61.4 | 56.1 | 49.6 | 54.9 | 84.4 | 63.8 |

Table 6: Ablation study on the effect of complete pruning at the late layer.

| | Cobra | | | | | RoboMamba | | | | |
|---|---|---|---|---|---|---|---|---|---|---|
| Condition | FLOPs | FLOPs Ratio | GQA | TextVQA | Vizwiz | FLOPs | FLOPs Ratio | GQA | POPE | OKVQA |
| w/o complete pruning | 1.26 | 100% | 61.5 | 56.1 | 49.6 | 0.44 | 100% | 55.1 | 84.4 | 63.9 |
| w/ complete pruning | 0.97 | 83% | 61.4 | 56.1 | 49.6 | 0.34 | 77% | 54.9 | 84.4 | 63.8 |

importance. Comparing the output term $y_t$, the input coefficient $B_t$, the state coefficient $C_t$, and the temporal delta term $\Delta_t$, it was found that $\Delta_t$ provides the most stable and effective signal in both Cobra and RoboMamba. The output term $y_t$ shows generally competitive performance but falls short of $\Delta_t$ on TextVQA, while $B_t$ and $C_t$ exhibit relatively lower performance, particularly on TextVQA for Cobra and on POPE for RoboMamba. In contrast, $\Delta_t$ achieves the highest or comparable scores across all datasets in the ablation study, demonstrating that it serves as the most reliable criterion for distinguishing salient from redundant tokens. These results justify our choice of adopting $\Delta_t$ as the default measure for token importance.

**Exploring strategies for token selection in pruning.** Table 5 compares three strategies for selecting pruning candidates. The first baseline simply selects tokens at random. The second strategy applies a Top-$k$ policy but also allows text tokens to be pruned. This strategy leads to catastrophic degradation across all benchmarks, with Cobra and RoboMamba both showing dramatic performance drops that make the results far below usable levels. These results highlight that text tokens are essential for reasoning and must not be removed. The third approach applies Top-$k$ only to visual tokens. This strategy reduces redundancy while maintaining overall performance stably, showing that visual tokens can be pruned safely without affecting model performance. These results support the validity of a ranked Top-$k$ pruning policy that targets only visual tokens.

**Effect of complete pruning at the late layer.** Table 6 reports the ablation results of complete pruning at the late layer, while keeping the selective pruning in the early layer fixed at a pruning ratio of 0.5. For Cobra, enabling complete pruning reduces FLOPs from 1.26 to 0.97, which corresponds to a 23% reduction, while accuracy remains nearly identical across GQA, TextVQA, and Vizwiz. Likewise, RoboMamba achieves a reduction from 0.44 to 0.34, amounting to a 23% decrease, with no meaningful accuracy drop on GQA, POPE, and OKVQA.These results emphasize that complete pruning at the late layer offers significant computational savings without sacrificing performance, validating it as an effective strategy for improving efficiency.

## 6 CONCLUSION

In this paper, we introduced Delta-guided Two stage Pruning (DTP), a novel framework for token pruning in Mamba-based MLLMs. Token importance is derived from the internal parameter $\Delta_t$ in Mamba's selective SSM, and the statistical distribution of these scores is analyzed together with implicit attention patterns to determine where pruning should occur and which tokens should be removed. Extensive experiments revealed that, under the same FLOPs budget, DTP preserves task performance more effectively than alternative pruning approaches. Ablation studies further demonstrated that $\Delta_t$ provides the most reliable criterion for estimating token importance, and that selecting the top-$k$ visual tokens based on these scores is a reasonable pruning strategy. Moreover, we confirmed that applying complete pruning at late layers maintains performance equivalent to retaining tokens at those layers. Taken together, these findings establish DTP as an effective method for pruning visual tokens in Mamba-based MLLMs.

## ACKNOWLEDGMENTS

This work was supported by Institute for Information & Communications Technology Planning & Evaluation (IITP) grant funded by the Korea government(MSIT) (No.RS-2025-25441838, Development of a human foundation model for human-centric universal artificial intelligence and training of personnel)

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

# A  TOKEN IMPORTANCE IN ROBOMAMBA

We provide further analyses on token importance using another Mamba-based MLLM, Robomamba. The figures show a similar trend to those observed in Cobra, supporting the generality of our pruning strategy.

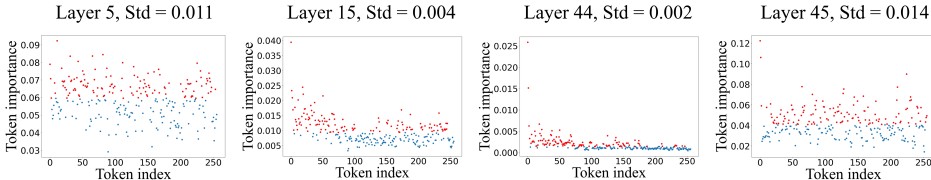

Figure 6: Visualization of token importance distributions at different layers of Robomamba (Liu et al., 2024). Red and blue points denote tokens within and outside the top 50% by importance, respectively.

# B  FULL LAYER-WISE STANDARD DEVIATION ANALYSIS

This section provides the complete layer-wise standard deviation plots for all datasets used in our evaluation. We sample 50 examples from each benchmark and compute the standard deviation of token-importance values across all 64 layers for both Cobra and RoboMamba. As shown in Figures 7 and 8, all datasets exhibit highly consistent trends. These results reinforce our findings in Section 4.2, demonstrating that layer-wise standard deviation serves as a reliable and architecture-agnostic indicator for identifying early and late pruning layers

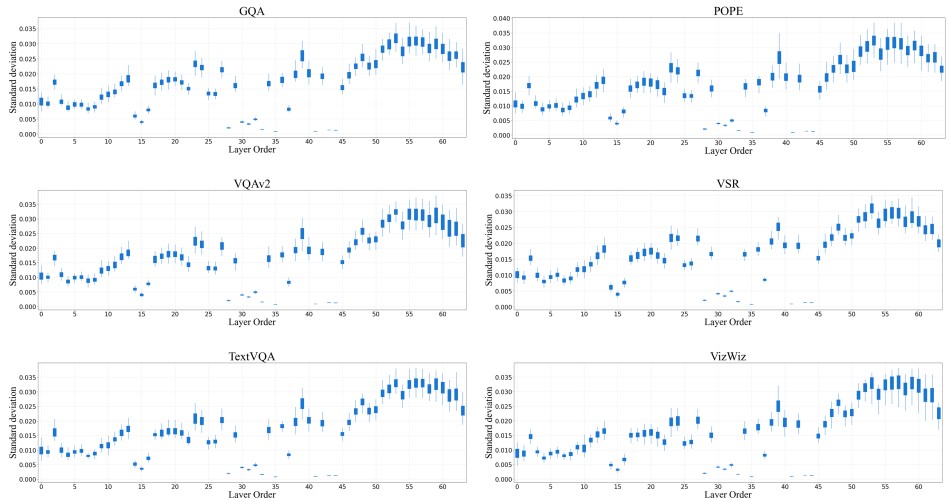

Figure 7: Layer-wise standard deviation of token importance values for the Cobra (Zhao et al., 2025) across all benchmark datasets.

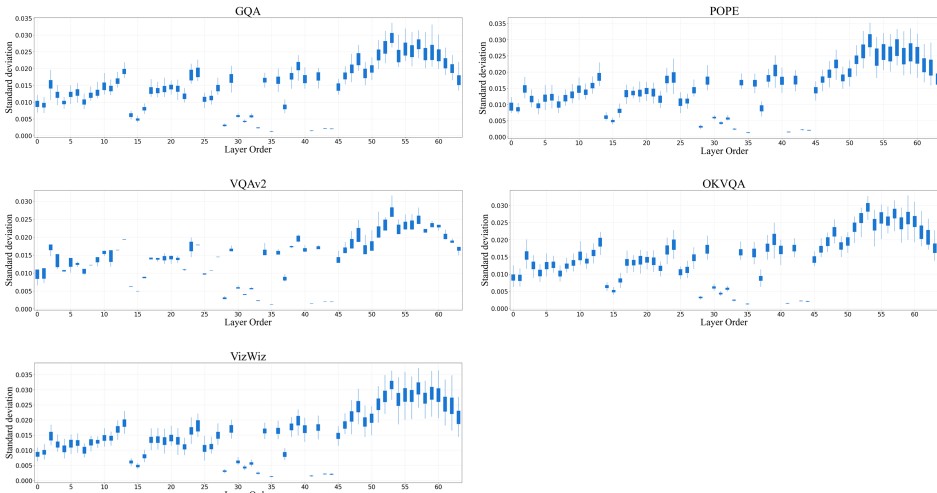

Figure 8: Layer-wise standard deviation of token importance values for the RoboMamba (Liu et al., 2024) across all benchmark datasets.

## C  FULL IMPLICIT ATTENTION PATTERNS

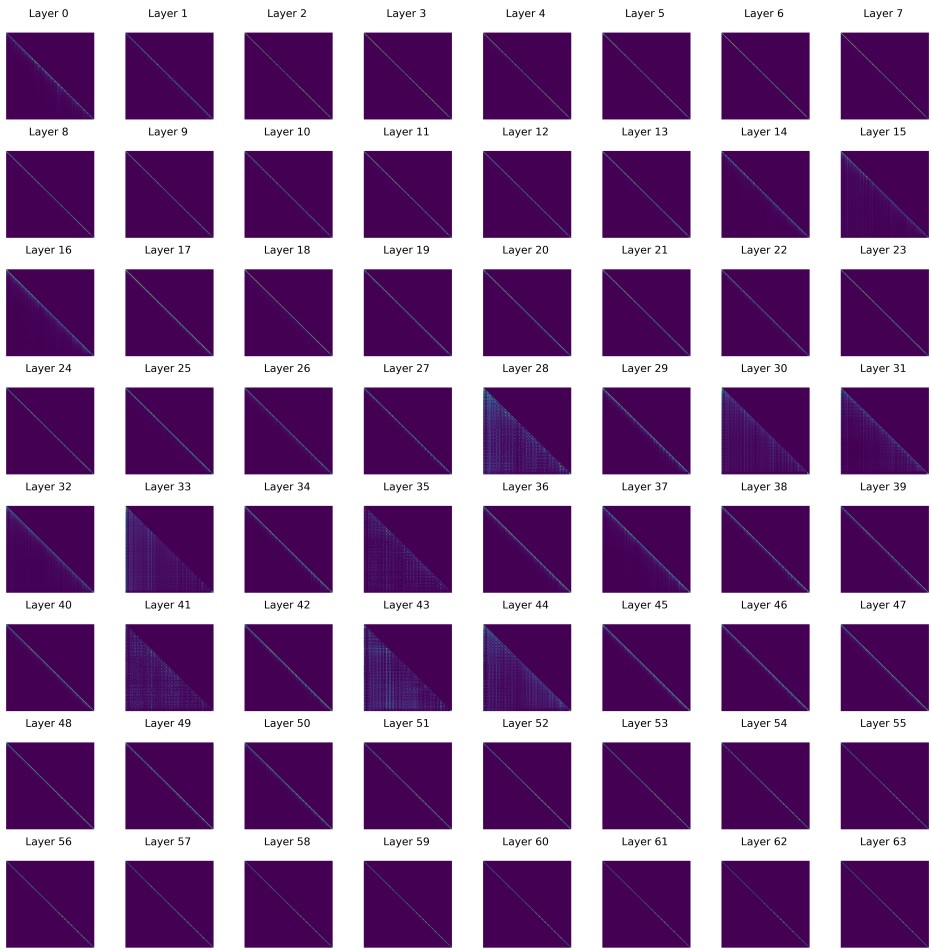

Figure 9: Full implicit attention patterns of each layer of Cobra.

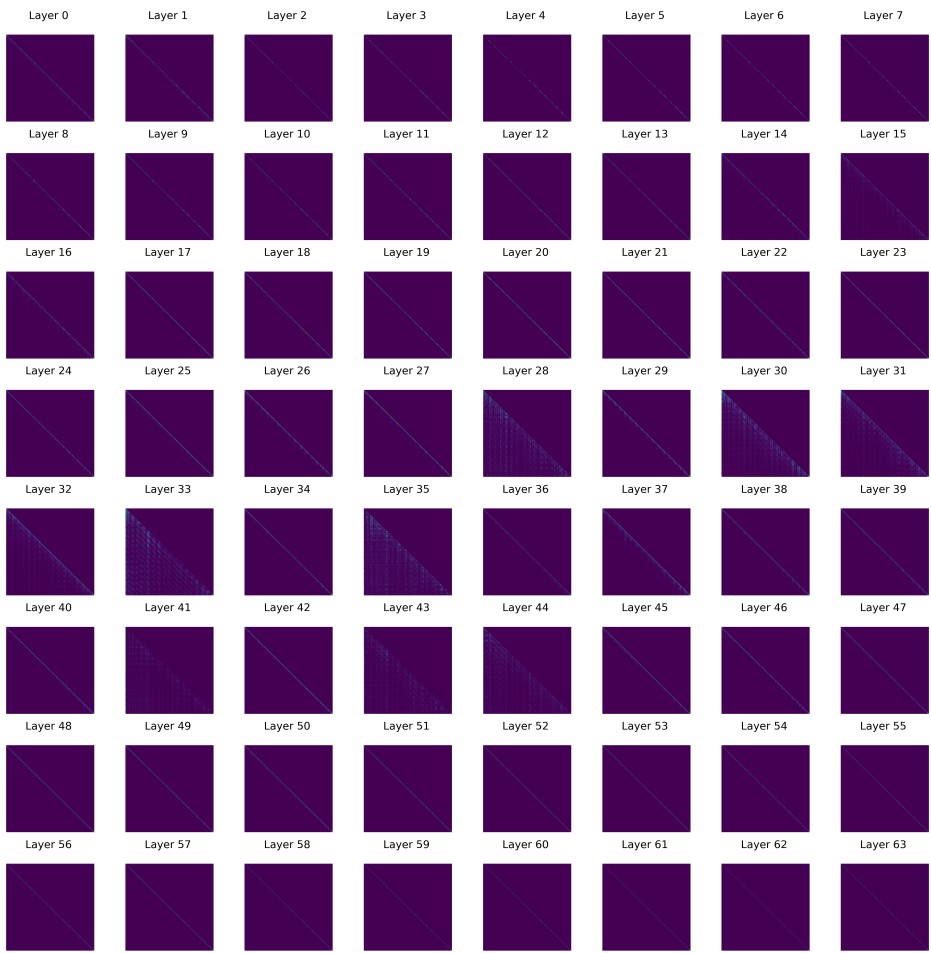

Figure 10: Full implicit attention patterns of each layer of RoboMamba.

# D    PERFORMANCE STABILITY ACROSS VARYING KEEP RATIOS

To examine how DTP behaves under different pruning strengths, we visualize the relationship between the keep ratio $r$ and the average score computed over all evaluation benchmarks. As shown in Figure 11, both Cobra and RoboMamba display smooth and monotonic performance degradation as $r$ decreases from 1.0 to 0.3, without any abrupt drops or signs of instability. This confirms that the method remains robust even when a large portion of visual tokens is removed.

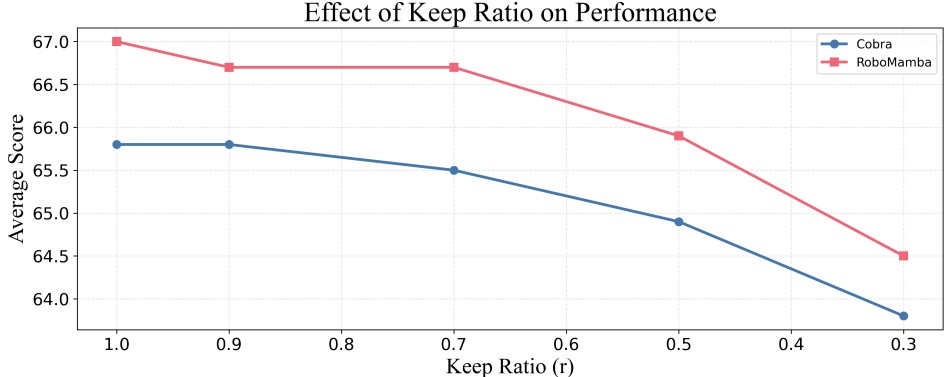

Figure 11: Performance trends across keep ratios ($r$).

