# OpenReview forum: "DTP: Delta-Guided Two Stage Pruning for Mamba-based Multimodal Large Language Models"
_ICLR.cc/2026/Conference — ICLR 2026 Poster_

### Official Review · Reviewer_pdx2 · 2025-10-25

**Soundness:** 2
**Presentation:** 3
**Contribution:** 2
**Rating:** 4
**Confidence:** 4

**Summary:**

This paper targets the inference-efficiency bottleneck of Mamba-based Multimodal Large Language Models (MLLMs) and proposes **Delta-guided Two-stage Pruning (DTP)**: early **top-k** retention of visual tokens and late **complete removal**, performed entirely at inference without retraining. Guided by the $\Delta_t$ based token importance, the authors select **layer 15** for selective pruning and **layer 45** for complete pruning. Experiments on **Cobra** and **RoboMamba** indicate that DTP can reduce FLOPs by **approximately 50%** while maintaining competitive accuracy.

**Strengths:**

1. **Architecture-aligned pruning signal:** Avoids modifying model structure or additional training; the target layers are determined via forward passes only.
2. **Two-stage design:** Preserves sufficient early visual information while removing later-layer redundancy, substantially improving efficiency.

**Weaknesses:**

1. **Methodology flaws:**
   a. The 15/45 choices hinge on layer-wise standard deviation of $\Delta_t$ -derived token importance computed on a **calibration dataset**; the authors mention a **VQAv2 subset** but disclose neither its size nor sampling strategy. Please detail these and validate on **multiple, distinct calibration datasets**.
   b. A natural **ablation by perturbing the pruning layers** is missing: plots show a $Std_\ell$ valley near **layer 15** and another around **layers 30–35**, yet no experiments/discussion evaluate these alternatives.
   c. Reporting the **post-pruning** standard-deviation profiles would further substantiate the effectiveness of the proposed selection.
2. **Experimental clarity and completeness:**
   a. In **Table 5**, clarify the comparison between *complete pruning* and *disable complete pruning*: in the latter, what pruning rate is used, or is it the **vanilla** model?
   b. It is recommended that the authors provide experimental results with different pruning rates to demonstrate the effectiveness of their approach; the main results only include (r=0.9) and (r=0.5).
   c. Beyond **FLOPs**, include **wall-clock latency** (with **prefill/decoding** breakdown) and **memory footprint**; the two-stage paradigm may affect prefill and decode differently even under the same global token compression.
   d. The authors should provide details such as the sampling parameters of the specific experiments to improve the reproducibility of the paper.

**Questions:**

Refer to Weakness

---

> ### Author Response · Authors · 2025-11-23
>
> We sincerely appreciate your careful reading and constructive feedback, which helped us clarify and strengthen several aspects of our method.
>
> -----
>
> ## Response to Weakness 1.a
>
> Thank you for the thoughtful suggestion. We agree that the calibration procedure should be described more clearly.
>
> We clarify that the calibration subset used to compute the layer-wise standard deviation of $\Delta_t$-guided token importance consists of 50 samples, sampled at random from the evaluation set.
>
> -----
> Regarding the request to validate the procedure on multiple and distinct calibration subsets, we will include additional analyses in the revised manuscript. For each evaluation dataset, we draw a separate 50-sample calibration subset and compute the corresponding layer-wise standard deviation curves.
>
> Across all benchmarks used to evaluate the method, and for both Cobra and RoboMamba, these curves consistently exhibit the same  pattern.
>
> This stability across datasets and models provides strong evidence that the adaptive pruning-layer selection is not tied to a specific subset and generalizes robustly.
>
> -----
> Because the same structure appears across all calibration subsets, this consistency also provides a strong foundation for formulating principled selection rules for the early and late pruning layers.
>
> In other words, the repeated emergence of these characteristic layer-wise standard deviation patterns naturally supports the formal criteria we introduce, which helps resolve the additional concern raised by other reviewers that the 15th and 45th layers may otherwise appear heuristic.
>
>
> So, we will refine the manuscript to present pruning-layer selection as a fully general and model-adaptive procedure rather than a fixed choice of layers.
>
> -----
> The revision will present the following generalizable procedure:
>
> **Selective pruning at early layer** is performed by locating the layer with the minimum standard deviation of $\Delta_t$-guided token importance within the first 25% of model depth. The validity of this criterion is further supported by the token importance distributions and implicit attention patterns, both of which consistently show clear clustering and strong token–token interactions near this region.
>
> $k_{\text{early}} = \arg\min_{\ell} Std_{\ell}$, where $0 \le \ell \le 0.25L$
>
> **Complete pruning at late layer**is performed by finding the layer that exhibits the largest inter-layer change in standard deviation within the final 30% of the depth. This point corresponds to the onset of the late layer cliff in the standard deviation curve. Beyond this region, implicit attention patterns show a marked weakening of token–token interactions, indicating that visual tokens no longer contribute stable or discriminative information to the computation. As a result, selective pruning becomes unreliable, and complete pruning becomes the more effective strategy.
>
> $k_{\text{late}} = \arg\max_{\ell} |Std_{\ell+1} - Std_{\ell}| + 1$, where $0.7L \le \ell < L-1$.
>
>
> -----
>
>
> We will incorporate these clarifications, together with results from multiple calibration subsets, into the revised manuscript and appendix.

---

> ### Author Response · Authors · 2025-11-23
>
> ## Response to Weakness 1.b
>
> Thank you for the helpful suggestion. To examine whether nearby pruning layers could also serve as reasonable choices, we conducted an ablation by perturbing both the early selective pruning layer and the late complete pruning layer around the points identified in our analysis. In our method, the optimal layers selected by the adaptive rules are $k_{\text{early}} = 15$ for selective pruning and $k_{\text{late}} = 45$ for complete pruning.
>
> -----
> For the early stage, selective pruning is applied at the layer where the standard deviation of the $\Delta_t$-guided token-importance scores reaches its minimum within the first 25% of the model depth. In the layer-wise standard-deviation curves, we observed that layers 14 and 16 exhibit similarly low standard-deviation values, comparable to layer 15.  We therefore evaluated these layers as perturbation candidates while keeping the late-stage pruning fixed at the optimal value $k_{\text{late}} = 45$
>
> The results show that, across both Cobra and RoboMamba, layer 15 consistently provides equally strong or best performance among the neighboring early layers. These observations indicate that the early layer low standard deviation region is broad but the most reliable and highest performing pruning point is centered at layer 15.
>
> -----
> For the late stage, complete pruning is applied at the layer showing the largest inter-layer change in standard deviation within the final 30% of model depth. During this analysis, we also identified layers 34 and 36 as locations where the absolute change in standard deviation is relatively large, even though they fall outside the designated late-collapse region. We therefore evaluated these layers as perturbation candidates while keeping the early-stage pruning fixed at the optimal value $k_{\text{early}} = 15$
>
> The results clearly show that pruning at layers 34 or 36 leads to substantial performance degradation.
> This reinforces the validity of our late-layer selection rule and demonstrates that reliable complete pruning is only achievable at the deeper point where the true cliff point begins.
>
> -----
>
> ### Cobra Results
> | Pruning Type       | Layer | GQA  | TextVQA | VizWiz |
> |--------------------|--------|------|---------|--------|
> | selective pruning  | 14     | 61.5 | 56.3    | 49.5   |
> |                    | 15     | 61.4 | 56.1    | 49.6   |
> |                    | 16     | 61.4 | 56.0    | 49.5   |
> | complete pruning   | 34     | 47.3 | 42.0    | 48.3   |
> |                    | 36     | 48.9 | 43.4    | 47.2   |
> |                    | 45     | 61.4 | 56.1    | 49.6   |
>
> ### RoboMamba Results
> | Pruning Type       | Layer | GQA  | POPE | OKVQA |
> |--------------------|--------|------|------|--------|
> | selective pruning  | 14     | 54.9 | 84.4 | 62.9   |
> |                    | 15     | 54.9 | 84.4 | 63.8   |
> |                    | 16     | 54.9 | 84.3 | 63.8   |
> | complete pruning   | 34     | 44.1 | 82.4 | 40.2   |
> |                    | 36     | 44.8 | 83.5 | 41.0   |
> |                    | 45     | 54.9 | 84.4 | 63.8   |

---

> ### Author Response · Authors · 2025-11-23
>
> ## Response to Weakness 1.c
>
> To examine the effect of pruning on the standard-deviation profiles, we computed the layer-wise standard deviation of the $\Delta_t$-guided token-importance scores both before and after pruning.
>
> -----
> We observed that, after the selective pruning at early layer, the standard deviation increases compared to the pre-pruning curve.
>
> This behavior is expected for the following reason:
>
> As shown in Figure 4 of the original draft, the low standard deviation region is characterized by a distribution where most tokens have similarly low importance values, and a few number of tokens exhibit noticeably higher importance.
> This makes the separation between redundant and informative tokens the clearest in this region, which is additional evidance that selective pruning is applied here.
>
> After pruning removes many of the low-importance tokens while retaining the relatively higher-importance ones, the distribution of remaining importance values becomes more spread out. Because the lower importance tokens originally made up the majority of the distribution, their removal reduces the concentration around the mean, which naturally leads to an increase in standard deviation.
>
>
> **This increase does not indicate degraded stability; rather, it confirms that pruning successfully retained the most informative tokens and removed the less informative ones.**
>
>
> After complete pruning at the late layer removes all visual tokens, only text tokens remain in the computation. Once the visual tokens are eliminated, the variance no longer reflects any information relevant to the pruning process, and therefore does not provide additional insight beyond the pre-pruning profiles.
>
> -----
> Overall, the post-pruning standard deviation profiles remain structurally aligned with the pre-pruning profiles. The observed increase in standard deviation simply reflects that pruning removed many low-importance tokens and preserved the higher-importance ones, confirming that the pruning acted on the intended region.

---

> ### Author Response · Authors · 2025-11-23
>
> ## Response to Weakness 2.a
>
> Thank you for the question and for pointing out the ambiguity in Table 5. We apologize for the lack of clarity in describing the comparison setting.
>
> We will clarify the setting used in Table 5. In the ablation study, “disable complete pruning” does not correspond to the vanilla model; instead, it refers to a configuration where the early-layer selective pruning is still applied with a fixed pruning ratio of 0.5, but the late-layer complete pruning is turned off.
>
> -----
> We will revise the manuscript to explicitly state that the “disable complete pruning” condition maintains the same early-layer pruning ratio (r = 0.5) and differs only in whether late-layer complete pruning is applied. We hope this resolves the confusion and clarifies the purpose of the ablation.

---

> ### Author Response · Authors · 2025-11-23
>
> ## Response to Weakness 2.b
>
> We appreciate your comments regarding the need to evaluate under a broader range of pruning rates. In the original submission, our analysis focused primarily on two representative settings, r = 0.9 and r = 0.5, which corresponded to mild and moderate pruning scenarios. While these baselines effectively demonstrated the robustness of our two-stage pruning strategy, we agree that a more comprehensive sweep over the keep ratio r can provide deeper insights into sensitivity and scalability.
>
> To address this, we conducted an expanded set of experiments on both Cobra and RoboMamba, covering a wider range of pruning ratios from 1.0 down to 0.3. This extended evaluation allows us to measure how performance gracefully degrades as more aggressive pruning is applied. The resulting computation–performance trade-off curves will be included in the appendix.
> As shown in the tables below, both models exhibit smooth and monotonic performance degradation, confirming that our pruning mechanism remains stable even under substantial token reduction.
> Notably, even at the most aggressive configuration (r = 0.3), the models preserve approximately 96% of their original accuracy, despite operating at only 39% of the baseline FLOPs.
>
> | Method            | FLOPs | FLOPs Ratio | GQA   | VQA-v2 | TextVQA | POPE  | VSR   | VizWiz | Avg   |
> |-------------------|-------|-------------|-------|--------|---------|-------|-------|--------|-------|
> | Cobra (r = 1.0)   | 2.01  | 100%        | 62.3  | 77.8   | 58.2    | 88.4  | 58.4  | 49.7   | 65.8  |
> | Cobra (r = 0.9)   | 1.35  | 67%         | 62.0  | 77.7   | 57.9    | 88.3  | 58.9  | 49.7   | 65.8  |
> | Cobra (r = 0.7)   | 1.16  | 58%         | 62.17 | 77.6   | 57.4    | 87.9  | 58.3  | 49.9   | 65.5  |
> | Cobra (r = 0.5)   | 0.97  | 48%         | 61.4  | 77.1   | 56.1    | 87.3  | 57.9  | 49.6   | 64.9  |
> | Cobra (r = 0.3)   | 0.79  | 39%         | 60.0  | 75.7   | 53.2    | 86.3  | 57.9  | 50.0   | 63.8  |
>
> | Method                | FLOPs | FLOPs Ratio | OK-VQA | GQA   | VQA-v2 | POPE  | VizWiz | Avg   |
> |-----------------------|-------|-------------|--------|-------|--------|-------|--------|-------|
> | RoboMamba (r = 1.0)   | 0.70  | 100%        | 64.4   | 56.4  | 74.9   | 85.1  | 54.2   | 67.0  |
> | RoboMamba (r = 0.9)   | 0.46  | 66%         | 64.2   | 56.1  | 74.7   | 85.2  | 53.2   | 66.7  |
> | RoboMamba (r = 0.7)   | 0.40  | 57%         | 64.4   | 56.0  | 74.4   | 85.0  | 53.5   | 66.7  |
> | RoboMamba (r = 0.5)   | 0.34  | 49%         | 63.8   | 54.9  | 73.6   | 84.4  | 52.8   | 65.9  |
> | RoboMamba (r = 0.3)   | 0.27  | 39%         | 63.4   | 52.9  | 71.74  | 82.78 | 51.7   | 64.5  |
>
> ----
> Overall, the analysis confirms that DTP operates reliably across a wide range of pruning strengths, demonstrating the practicality and robustness of the proposed method for Mamba-based MLLMs.

---

> ### Author Response · Authors · 2025-11-23
>
> ## Response to Weakness 2.c
>
> We appreciate your thoughtful suggestion to report wall-clock latency and memory footprint.
> In response, we conducted additional experiments on the Cobra model using the POPE dataset to measure prefill latency, decode latency, total latency, and GPU memory usage.
>
> | Method            | FLOPs | Prefill Mean (ms) | Decode Mean (ms) | Total Latency | GPU Memory |
> |-------------------|-------|-------------------|-------------------------|----------------|------------|
> | Cobra (Original)  | 2.01  | 98.04             | 5.04                    | 16m 05s        | 8.8 GB     |
> | FastV (k=2, r=0.5)       | 1.06  | 60.64             | 4.94                | 10m 24s        | 8.5 GB     |
> | VTW (k=32)          | 1.04  | 67.31             | 4.99                    | 11m 25s        | 8.3 GB |
> | DART (r=0.44)        | 0.96  | 76.45             | 4.95                    | 12m 44s        | 8.5 GB     |
> | DTP (r=0.5)     | 0.97  | 61.54             | 5.02                    | 10m 35s        | 8.3 GB     |
>
> The results show that our proposed method, DTP (r = 0.5), substantially reduces the prefill latency from 98.04 ms to 61.54 ms and decreases the total inference time from 16m 05s to 10m 35s, achieving an approximate 40% real-world speedup. GPU memory usage is also reduced compared to the original model. Notably, these improvements are competitive with, and in many cases superior to, existing token-pruning approaches such as FastV, VTW, and DART [1].
> We also note that DART [1] was included as an additional pruning method to expand the baseline set in response to another reviewer’s request. DART [1] prunes tokens through pivot selection and cosine similarity rather than attention scores, making it architecture-agnostic and directly applicable to Mamba-based models.
>
> Although decode latency does not change significantly, this behavior is expected and fully consistent with the computational characteristics of Mamba-based MLLMs, as described in Figure 1 and lines 50–51 of the original draft. Specifically, the prefill stage involves computation over the entire input sequence, whereas the decode stage generates each new token using only the previous hidden state and the most recently generated token. As a result, pruning has inherently limited influence on decode stage latency, and the decode time cost therefore remains essentially unchanged.
> Moreover, more than 95% of the total computational cost in Mamba-based MLLMs is incurred during the prefill stage, whereas the decode stage accounts for only a very small fraction of the overall latency. Consequently, even if decode latency were improved, its contribution to the total inference efficiency would be marginal compared to the dominant prefill cost.
>
> These inference latency results will be incorporated into the revised manuscript in Section 5.3 (Efficiency Analysis), clearly demonstrating that the FLOPs reduction achieved by DTP directly translates into substantial wall-clock performance improvement.
>
> -----
> [1] Wen, Zichen, et al. "Stop looking for important tokens in multimodal language models: Duplication matters more." arXiv preprint arXiv:2502.11494 (2025).

---

> ### Author Response · Authors · 2025-11-23
>
> ## Response to Weakness 2.d
>
> Thank you for pointing out the importance of clearly reporting the sampling parameters used in our experiments to ensure reproducibility.
>
> All calibration subsets used to compute layer-wise standard deviation curves were constructed by 50 samples, sampled at random from each evaluation dataset. This sampling strategy was applied consistently across all benchmarks and models.
>
> If there are any additional experimental details that require explicit clarification, we will make sure to incorporate them during the manuscript revision process so that the full procedure is transparent and fully reproducible.
>
> -----
> Thank you once again for your valuable feedback. Your comments have allowed us to perform additional analyses and clarifications that have meaningfully strengthened the paper, and we will incorporate all corresponding updates in the revised manuscript as promptly as possible.
>
> We hope that our responses adequately address the concerns and questions you raised, and we would greatly appreciate it if you could kindly re-evaluate the work based on the explanations and results provided here.
>
> If there are any further points you would like to discuss or additional suggestions you wish to offer, please feel free to let us know at any time. Your feedback is always welcome, and we remain committed to continuously improving this research based on your insightful comments.

---

> > ### Comment · Reviewer_pdx2 · 2025-11-25
> >
> > Thank you for the author's response. The author's response has resolved most of my questions, and I will improve my score.

---

> > > ### Author Response · Authors · 2025-11-26
> > >
> > > We are very glad to hear that our answers resolved most of your questions!
> > >
> > > Your constructive suggestions have greatly helped us refine and strengthen our work.
> > >
> > > We sincerely appreciate the time and effort you devoted to providing thoughtful feedback.
> > > Thank you again for your valuable comments.

---

### Official Review · Reviewer_oGU1 · 2025-10-26

**Soundness:** 3
**Presentation:** 3
**Contribution:** 3
**Rating:** 6
**Confidence:** 4

**Summary:**

This paper introduces DTP (Delta-guided Two stage Pruning), a novel and effective token pruning framework designed for Mamba-based Multimodal Large Language Models (MLLMs) to reduce the high computational cost of the prefill stage, which is dominated by a large number of visual tokens. The core idea is to leverage an internal, input-dependent parameter of the Mamba architecture, ∆t, to estimate the importance of each visual token without requiring any retraining. The proposed DTP method employs a two-stage strategy, beginning with selective pruning at an early layer (the 15th), where a portion of the least important visual tokens are discarded. This is followed by complete pruning at a late layer (the 45th), where all remaining visual tokens are removed, a decision justified by the observation that their contributions become negligible in deeper layers as implicit attention patterns diminish. Extensive experiments on two Mamba-based MLLMs (Cobra and RoboMamba) demonstrate that DTP can reduce computation (FLOPs) by nearly 50% while incurring minimal performance degradation, significantly outperforming adapted Transformer-based pruning methods.

**Strengths:**

- The paper addresses a relevant and important problem: the inference inefficiency of Mamba-based MLLMs. It is intrinsically tied to the Mamba architecture by using the ∆t parameter for importance scoring, a concept not applicable to Transformers.
- The design choices, particularly the selection of the 15th and 45th layers for pruning, are not made ad-hoc. They are convincingly supported by a careful analysis of the model's internal state, which adds a layer of interpretability and principle to the method.
- The method achieves a remarkable balance between computational reduction and performance preservation. A nearly 50% reduction in FLOPs with an average performance drop of less than 1 point (on the Cobra model) is a very strong result. The comprehensive ablation studies further solidify the paper's claims.

**Weaknesses:**

- The specific layers for pruning (15th and 45th) were empirically identified for the Cobra and RoboMamba models, which appear to have around 64 layers. It is unclear how these specific layer indices would generalize to Mamba-based models of different depths (e.g., a 48-layer or 96-layer model). While the methodology for finding these layers (analyzing the standard deviation of ∆t) seems general, the paper could be strengthened by framing it as a general "recipe" and discussing how it would apply to other architectures.
- The evaluation relies exclusively on FLOPs as a metric for computational cost. While FLOPs are a good hardware-agnostic proxy, they do not always translate linearly to wall-clock speedup due to factors like memory access patterns and GPU kernel optimizations. Including actual latency measurements ( ms/token or similar) would provide a more practical and complete picture of the efficiency gains.
- The experiments show results for keep ratios r of 0.9 and 0.5. A more detailed analysis of the trade-off between performance and the keep ratio r would be beneficial. A plot showing how the average performance gracefully degrades as r is decreased would give readers a better sense of the method's sensitivity to this hyperparameter.
- It is recommended to include a comparison and discussion with these methods [1-4].

[1] Arif, Kazi Hasan Ibn, et al. "HiRED: Attention-Guided Token Dropping for Efficient Inference of High-Resolution Vision-Language Models." Proceedings of the AAAI Conference on Artificial Intelligence. Vol. 39. No. 2. 2025. \
[2] Xing, Long, et al. "Pyramiddrop: Accelerating your large vision-language models via pyramid visual redundancy reduction." arXiv preprint arXiv:2410.17247 (2024).  \
[3] Wen, Zichen, et al. "Stop looking for important tokens in multimodal language models: Duplication matters more." arXiv preprint arXiv:2502.11494 (2025).  \
[4] Ye, Weihao, et al. "Fit and prune: Fast and training-free visual token pruning for multi-modal large language models." Proceedings of the AAAI Conference on Artificial Intelligence. Vol. 39. No. 21. 2025.

**Questions:**

- Regarding the choice of the 15th and 45th layers: Could you elaborate on the generality of this finding? If a researcher were to apply DTP to a new Mamba-based MLLM with a different number of layers, should they re-run the standard deviation analysis to find the new optimal "early" and "late" layers? Is there a rule of thumb, e.g., pruning at \~25% and \~70% of the model's depth?
- The paper's efficiency claims are based on FLOPs reduction. Have you conducted any experiments to measure the actual wall-clock inference speedup (e.g., in terms of tokens/second or total latency)? This would be a valuable addition to confirm the practical benefits of DTP.
- How sensitive is the DTP framework to the choice of the early-stage keep ratio r? Could you provide a brief analysis or a curve illustrating the performance-computation trade-off as r is varied continuously between, for example, 0.5 and 1.0?

---

> ### Author Response · Authors · 2025-11-21
>
> We sincerely appreciate your thoughtful comment and provide the following response to fully address the concern.
>
> ----
> ## Response to Weakness 1
>
> We fully agree that presenting our method as a general “recipe” rather than reporting fixed layer indices can make the framework more broadly applicable to Mamba-based models of varying depths.
>
> In the revised manuscript, we will clarify that the 15th and 45th layers reported for Cobra and RoboMamba are not fixed heuristics, but empirical instances of a general two-stage pruning procedure derived from Section 4.2.
>
> ----
> The updated section now highlights the following generalizable procedure:
>
> **Selective pruning at early layer** is performed by locating the layer with the minimum standard deviation of $\Delta_t$-guided token importance within the first 25% of model depth. The validity of this criterion is further supported by the token importance distributions and implicit attention patterns, both of which consistently show clear clustering and strong token–token interactions near this region.
>
> $k_{\text{early}} = \arg\min_{\ell} Std_{\ell}$, where $0 \le \ell \le 0.25L$
>
> ----
> **Complete pruning at late layer**is performed by finding the layer that exhibits the largest inter-layer change in standard deviation within the final 30% of the depth. This point corresponds to the onset of the late layer cliff in the standard deviation curve. Beyond this region, implicit attention patterns show a marked weakening of token–token interactions, indicating that visual tokens no longer contribute stable or discriminative information to the computation. As a result, selective pruning becomes unreliable, and complete pruning becomes the more effective strategy.
>
> $k_{\text{late}} = \arg\max_{\ell} |Std_{\ell+1} - Std_{\ell}| + 1$, where $0.7L \le \ell < L-1$.
>
> ----
> Across all benchmark datasets and for both Cobra and RoboMamba, we consistently observed the same patterns of layer-wise standard deviation, providing strong empirical support that these criteria generalize well beyond the specific models studied. We will further supplement the manuscript by adding the layer-wise standard deviation plots in the appendix, ensuring that these observations are transparently documented.
> We believe this clarification directly addresses your concern and improves the clarity and general applicability of our method.

---

> ### Author Response · Authors · 2025-11-21
>
> ## Response to Weakness 4 (1/2)
> We sincerely appreciate your suggestion to include comparisons with the recent token-pruning methods [1–4]. As noted in lines 77–82 of the manuscript, most existing approaches are designed under the assumptions of Transformer architectures, and these structural dependencies make them difficult to apply directly to Mamba-based models. This architectural mismatch was, in fact, one of the motivations behind our work and led us to develop a new pruning strategy tailored specifically to the characteristics of Mamba. In the following response, we provide an integrated discussion focusing on whether each method can be meaningfully applied to Mamba-based Multimodal LLMs.
>
> -----
> **HiRED [1]** is a token dropping method tailored for the Transformer-based high-resolution Vision-Language Model LLaVA-Next. It reduces the large number of visual tokens generated during high-resolution image processing by leveraging CLS-attention to select informative tokens and allocate an appropriate token budget. However, HiRED operates as an early dropping strategy, removing tokens at the vision-encoder stage before the multimodal reasoning process begins. In contrast, our approach performs pruning inside the Mamba-based MLLM, targeting the internal multimodal representations rather than the visual encoding stage. Due to this fundamental difference in where and how pruning is applied, HiRED does not serve as an appropriate comparison baseline for our work.
>
> **PyramidDrop [2]** observes that all visual tokens are necessary in shallow layers, whereas redundant tokens increasingly emerge in deeper layers, and it removes a predefined proportion of image tokens at each stage accordingly. To determine which tokens should be discarded, the method leverages the Transformer's self-attention mechanism by computing the similarity between the query state of the last instruction token and the key states of all image tokens, treating this similarity as a measure of image-token importance. In other words, PyramidDrop evaluates the visual tokens based on how strongly they are attended by the instruction token, which inherently relies on the explicit query–key projections and token-level attention interactions available in Transformer architectures. However, Mamba does not employ Q/K/V projections and instead processes the input sequence through a recurrent state-space scan that incrementally compresses information into a hidden state, making such text–image interaction signals impossible to replicate. For these reasons, PyramidDrop is structurally incompatible with Mamba-based MLLMs and is therefore not an appropriate comparison baseline for our study.
>
> **FitPrune [4]** formulates token pruning as preserving both self-attention and cross-attention distributions so that the divergence between pre- and post-pruning attention patterns is minimized. However, this approach fundamentally assumes the presence of explicit attention matrices provided by Transformer architectures, whereas Mamba does not generate such attention maps. In our work, we analyze Mamba’s implicit attention behavior, but this serves only as a qualitative tool to understand where token interactions tend to occur and is not directly used as a pruning criterion. In particular, the cross-attention distribution required by FitPrune cannot be computed in Mamba because the model lacks explicit attention and instead processes the input through a recurrent scan that compresses information into a hidden state, making token-to-token comparisons fundamentally inaccessible. This architectural incompatibility makes direct comparison with FitPrune inappropriate. Our pruning strategy is instead built around the ∆t parameter, which reflects Mamba’s input-dependent selective update mechanism. This represents a fundamentally different design philosophy from Transformer-based pruning methods that rely on explicit attention statistics.
>
> **DART [3]** does not evaluate token importance through attention scores, unlike the other methods, and is designed to remove redundant visual tokens by selecting pivot tokens and using cosine similarity with other tokens. Therefore, it can operate in a similar manner not only in Transformer architectures but also in Mamba-based models, and we consider it a meaningful method that can be compared in our setting.
>
> ----
> Accordingly, we included DART [4] as a comparison baseline and evaluated performance on both Cobra and RoboMamba models under comparable FLOPs settings. The experimental results are shown in the table below, and DTP consistently achieves the best performance, further clarifying the effectiveness of our proposed approach. We will add these results to Section 5.2 (MAIN RESULTS) of the revised manuscript.

---

> ### Author Response · Authors · 2025-11-21
>
> ## Response to Weakness 4 (2/2)
>
> | Method                  | FLOPs | FLOPs Ratio | GQA  | VQA-v2 | TextVQA | POPE | VSR  | VizWiz | Avg  |
> |-------------------------|-------|-------------|------|--------|---------|------|------|--------|------|
> | Cobra                   | 2.01  | 100%        | 62.3 | 77.8   | 58.2    | 88.4 | 58.4 | 49.7   | 65.8 |
> |         |       |             |      |        |         |      |      |        |      |
> | FastV (k=2, r=0.7)      | 1.45  | 72%         | 62.1 | 77.4   | 56.9    | 87.7 | 58.0 | 49.8   | 65.3 |
> | VTW (K=45)              | 1.43  | 71%         | 62.1 | 77.7   | 58.2    | 88.3 | 58.5 | 49.5   | 65.7 |
> | DART (r=0.66)           | 1.38  | 68.7%       | 62.0 | 77.1   | 57.0    | 87.4 | 58.2 | 49.7   | 65.2 |
> | DTP (r=0.9)             | 1.35  | 67%         | 62.0 | 77.7   | 57.9    | 88.3 | 58.9 | 49.7   | 65.8 |
> |         |       |             |      |        |         |      |      |        |      |
> | FastV (k=2, r=0.5)      | 1.06  | 53%         | 61.7 | 76.8   | 55.0    | 87.4 | 57.3 | 50.1   | 64.7 |
> | VTW (K=32)              | 1.04  | 52%         | 47.1 | 54.1   | 42.6    | 74.1 | 57.9 | 48.5   | 54.0 |
> | DART (r=0.44)           | 0.96  | 48%         | 61.2 | 76.1   | 55.1    | 86.3 | 57.7 | 49.5   | 64.3 |
> | DTP (r=0.5)             | 0.97  | 48%         | 61.4 | 77.1   | 56.1    | 87.3 | 57.9 | 49.6   | 64.9 |
>
> | Method                  | FLOPs | FLOPs Ratio | OK-VQA | GQA  | VQA-v2 | POPE | VizWiz | Avg  |
> |-------------------------|-------|-------------|--------|------|--------|------|--------|------|
> | RoboMamba               | 0.70  | 100%        | 64.4   | 56.4 | 74.9   | 85.2 | 54.2   | 67.0 |
> |         |       |             |      |        |         |      |      |        |      |
> | FastV (k=2, r=0.7)      | 0.50  | 71%         | 64.1   | 56.1 | 74.4   | 85.3 | 53.0   | 66.6 |
> | VTW (K=45)              | 0.50  | 71%         | 64.0   | 55.9 | 74.7   | 85.3 | 53.3   | 66.6 |
> | DART (r=0.63)           | 0.47  | 67%         | 64.2   | 55.4 | 73.9   | 84.2 | 53.0   | 66.1 |
> | DTP (r=0.9)             | 0.46  | 66%         | 64.2   | 56.1 | 74.7   | 85.2 | 53.2   | 66.7 |
> |         |       |             |      |        |         |      |      |        |      |
> | FastV (k=2, r=0.5)      | 0.37  | 53%         | 63.4   | 55.1 | 73.4   | 84.2 | 52.1   | 65.6 |
> | VTW (K=32)              | 0.36  | 51%         | 40.0   | 44.7 | 55.2   | 82.1 | 45.2   | 53.4 |
> | DART (r=0.47)           | 0.36  | 51%         | 64.1   | 54.0 | 72.8   | 83.4 | 52.6   | 65.4 |
> | DTP (r=0.5)             | 0.34  | 49%         | 63.8   | 54.9 | 73.6   | 84.4 | 52.8   | 65.9 |

---

> ### Author Response · Authors · 2025-11-21
>
> ## Response to Weakness 2
> We truly appreciate your insightful remark about the limitations of relying only on FLOPs as a measure of computational efficiency. In response, we conducted additional experiments on the Cobra model using the POPE to measure prefill latency, decode latency, total latency, and GPU memory usage.
>
> | Method            | FLOPs | Prefill Mean (ms) | Decode Mean (ms) | Total Latency | GPU Memory |
> |-------------------|-------|-------------------|-------------------------|----------------|------------|
> | Cobra (Original)  | 2.01  | 98.04             | 5.04                    | 16m 05s        | 8.8 GB     |
> | FastV (k=2, r=0.5)       | 1.06  | 60.64             | 4.94                | 10m 24s        | 8.5 GB     |
> | VTW (k=32)          | 1.04  | 67.31             | 4.99                    | 11m 25s        | 8.3 GB |
> | DART (r=0.44)        | 0.96  | 76.45             | 4.95                    | 12m 44s        | 8.5 GB     |
> | DTP (r = 0.5)     | 0.97  | 61.54             | 5.02                    | 10m 35s        | 8.3 GB     |
>
> The results show that our proposed method, DTP (r = 0.5), substantially reduces the prefill latency from 98.04 ms to 61.54 ms and decreases the total inference time from 16m 05s to 10m 35s, achieving an approximate 40% real-world speedup. GPU memory usage is also reduced compared to the original model. Notably, these improvements are competitive with, and in many cases superior to, existing token-pruning approaches such as FastV, VTW, and DART.
>
> Although decode latency does not change significantly, this behavior is expected and fully consistent with the computational characteristics of Mamba-based MLLMs, as described in Figure 1 and lines 50–51 of the original draft. Specifically, the prefill stage involves computation over the entire input sequence, whereas the decode stage generates each new token using only the previous hidden state and the most recently generated token. As a result, pruning has inherently limited influence on decode stage latency, and the decode time cost therefore remains essentially unchanged.
> Moreover, more than 95% of the total computational cost in Mamba-based MLLMs is incurred during the prefill stage, whereas the decode stage accounts for only a very small fraction of the overall latency. Consequently, even if decode latency were improved, its contribution to the total inference efficiency would be marginal compared to the dominant prefill cost.
>
> ----
> These real-world inference latency results will be incorporated into the revised manuscript in Section 5.3 (Efficiency Analysis), clearly demonstrating that the FLOPs reduction achieved by DTP directly translates into substantial wall-clock performance improvement.

---

> ### Author Response · Authors · 2025-11-21
>
> ## Response to Weakness 3
> Thank you for pointing out the need for a broader evaluation across different pruning ratios. In the original submission, our analysis focused primarily on two representative settings, r = 0.9 and r = 0.5, which corresponded to mild and moderate pruning scenarios. While these baselines effectively demonstrated the robustness of our two-stage pruning strategy, we agree that a more comprehensive sweep over the keep ratio rcan provide deeper insights into sensitivity and scalability.
>
> ----
> To address this, we conducted an expanded set of experiments on both Cobra and RoboMamba, covering a wider range of pruning ratios from 1.0 down to 0.3. This extended evaluation allows us to measure how performance gracefully degrades as more aggressive pruning is applied. The resulting computation–performance trade-off curves will be included in the appendix. As shown in the tables below, both models exhibit smooth and monotonic performance degradation, confirming that our pruning mechanism remains stable even under substantial token reduction. Notably, even at the most aggressive configuration (r = 0.3), the models preserve approximately 96% of their original accuracy, despite operating at only 39% of the baseline FLOPs. This highlights the resilience of our approach, even in highly compressed regimes.
>
> | Method            | FLOPs | FLOPs Ratio | GQA   | VQA-v2 | TextVQA | POPE  | VSR   | VizWiz | Avg   |
> |-------------------|-------|-------------|-------|--------|---------|-------|-------|--------|-------|
> | Cobra (r = 1.0)   | 2.01  | 100%        | 62.3  | 77.8   | 58.2    | 88.4  | 58.4  | 49.7   | 65.8  |
> | Cobra (r = 0.9)   | 1.35  | 67%         | 62.0  | 77.7   | 57.9    | 88.3  | 58.9  | 49.7   | 65.8  |
> | Cobra (r = 0.7)   | 1.16  | 58%         | 62.2 | 77.6   | 57.4    | 87.9  | 58.3  | 49.9   | 65.5  |
> | Cobra (r = 0.5)   | 0.97  | 48%         | 61.4  | 77.1   | 56.1    | 87.3  | 57.9  | 49.6   | 64.9  |
> | Cobra (r = 0.3)   | 0.79  | 39%         | 60.0  | 75.7   | 53.2    | 86.3  | 57.9  | 50.0   | 63.8  |
>
> | Method                | FLOPs | FLOPs Ratio | OK-VQA | GQA   | VQA-v2 | POPE  | VizWiz | Avg   |
> |-----------------------|-------|-------------|--------|-------|--------|-------|--------|-------|
> | RoboMamba (r = 1.0)   | 0.70  | 100%        | 64.4   | 56.4  | 74.9   | 85.1  | 54.2   | 67.0  |
> | RoboMamba (r = 0.9)   | 0.46  | 66%         | 64.2   | 56.1  | 74.7   | 85.2  | 53.2   | 66.7  |
> | RoboMamba (r = 0.7)   | 0.40  | 57%         | 64.4   | 56.0  | 74.4   | 85.0  | 53.5   | 66.7  |
> | RoboMamba (r = 0.5)   | 0.34  | 49%         | 63.8   | 54.9  | 73.6   | 84.4  | 52.8   | 65.9  |
> | RoboMamba (r = 0.3)   | 0.27  | 39%         | 63.4   | 52.9  | 71.7  | 82.8 | 51.7   | 64.5  |
>
> ----
> Overall, the analysis confirms that DTP operates reliably across a wide range of pruning strengths, demonstrating the practicality and robustness of the proposed method for Mamba-based MLLMs.

---

> ### Author Response · Authors · 2025-11-21
>
> # Response to Question 1 to 3
>
> -----
>  ## Q1 (Related to Weakness 1)
> Thank you for your valuable question.
>
> In our method, the pruning layers are not fixed heuristics tied to specific layer indices. Instead, they are determined based on statistical patterns observed in the early region (approximately the first 25% of depth) and the late region (around 70% of depth) of the model.
>
> Specifically, for early layer pruning, the point where the standard deviation is minimal in the region is selected, and for late layer pruning, the point where the standard deviation change rate in the region increases the most is formulated and ultimately selected.
>
> Therefore, the 15th and 45th layers reported in the paper are not fixed rules but empirical outcomes obtained when applying this generalized selection procedure to Cobra and RoboMamba.
>
> When applying DTP to a new Mamba-based MLLM with a different number of layers, standard deviation analysis must be performed first.
> This analysis is performed on a subsample of the test data, and in practical applications, it is computationally inexpensive and simple to perform.
>
> Moreover, across multiple evaluation datasets and for both Cobra and RoboMamba, we consistently observed the same pattern, which strongly suggests that the proposed layer selection strategy is both reliable and broadly generalizable.
>
> -----
>  ## Q2 (Related to Weakness 2)
> We conducted additional experiments to measure prefill latency, decode latency, total inference time, and GPU memory usage on the Cobra model using the POPE benchmark.
>
> The results show that DTP (r = 0.5) reduces prefill latency from 98.04 ms → 61.54 ms and total inference time from 16m 05s → 10m 35s, achieving an approximate 40% wall-clock speedup, while also reducing GPU memory usage (8.8 GB → 8.3 GB).
>
> These latency analyses will be incorporated into Section 5.3 of the revised manuscript.
>
> -----
>  ## Q3 (Related to Weakness 3)
> We perform a wide range of pruning ratios (r = 1.0, 0.9, 0.7, 0.5, 0.3) on both Cobra and RoboMamba.
>
> For the Cobra, the average score decreased from 65.8 to 64.9 as r decreased from 1.0 to 0.5, while maintaining approximately 96% of the overall accuracy even at r = 0.3. RoboMamba showed a similar trend, maintaining approximately 96% of its performance even at r = 0.3.
>
> These results demonstrate that DTP is robust across a wide range of retention rates, and we will include performance-computation tradeoff curves in the appendix to illustrate this characteristic.
>
> ----
> Thank you once again for your valuable feedback. Your comments have enabled us to conduct additional analyses and experiments, which have strengthened the paper in several meaningful ways. We will update the revised manuscript as promptly as possible.
>
> We hope that our responses adequately address the concerns and questions you raised, and we would greatly appreciate it if you could kindly re-evaluate the work based on the explanations and results provided here.
>
> If there are any further points you would like to discuss or additional suggestions you wish to offer, please feel free to let us know at any time. Your feedback is always welcome, and we remain committed to continuously improving this research based on your insightful comments.

---

### Official Review · Reviewer_BGBy · 2025-10-30

**Soundness:** 3
**Presentation:** 3
**Contribution:** 3
**Rating:** 4
**Confidence:** 4

**Summary:**

This paper proposes Delta-guided Two Stage Pruning (DTP), a pruning framework designed for Mamba-based multimodal large language models (MLLMs). The method estimates token importance using the Mamba-specific internal parameter ∆t, enabling selective pruning in early layers and complete pruning in late layers. Experiments on Cobra and RoboMamba demonstrate that DTP can reduce FLOPs by nearly 50% with minimal accuracy degradation. The study further analyzes implicit attention patterns in Mamba, providing insight into token dynamics across layers.

**Strengths:**

- Proposes a new pruning paradigm specifically designed for Mamba-based models rather than Transformer-based ones.
- Experiments are comprehensive, covering both Cobra and RoboMamba across multiple benchmarks.
- The two-stage pruning strategy is intuitively reasonable and empirically validated.
- Provides interesting analysis of implicit attention patterns in Mamba, offering new perspectives on token behavior.

**Weaknesses:**

- Limited theoretical justification for ∆t as a universal token importance indicator; the claim is mostly empirical.
- The computational overhead of computing ∆t during inference is not discussed; clarity on latency gain beyond FLOPs would be beneficial.
- Comparison with non-pruning efficiency techniques (e.g., token merging, KV cache optimization) is missing.
- Some figures and analysis (e.g., implicit attention) could be better interpreted to help readers grasp the practical implications.

If authors address my concerns, I will consider raising my score.

**Questions:**

- Can ∆t be efficiently extracted in real inference pipelines without additional latency?

- How sensitive is DTP to the choice of pruning layers (15th and 45th)? Would adaptive layer selection improve performance?

- Could the ∆t-based importance be combined with other statistics (e.g., gradient-based) to further enhance robustness?

- How does DTP perform under extremely high pruning ratios (>60%)?

- Are the observations about implicit attention patterns consistent across all Mamba-based architectures?

---

> ### Author Response · Authors · 2025-11-22
>
> We sincerely appreciate your thoughtful and constructive comments. Your feedback provides a valuable opportunity to further improve the quality of our work, and we are committed to addressing your points with the utmost care and clarity.
>
> -----
> ## Response to Weakness 1
> Thank you for pointing out the need for a clearer theoretical justification. In the revised manuscript, we will update Section 4.1 to explicitly describe the structural motivation for using $\Delta_t$ as a token importance indicator.
>
> The updated section will include the following explanation:
> >
> >“Specifically, $\Delta_t$ directly modulates both $\bar{A}_t$ and $\bar{B}_t$, thereby controlling the discretized state transition dynamics and determining the extent to which each token contributes to subsequent hidden states. Tokens with larger $\Delta_t$ values induce stronger state updates, whereas those with smaller values yield only minor transitions, effectively acting as less influential inputs.”
>
> Through this explanation, we will clarify that $\Delta_t$ directly regulates both $\bar{A}_t$ and $\bar{B}_t$, thereby determining the magnitude of the discretized state transitions. This naturally positions $\Delta_t$ as a structurally grounded measure of token importance in Mamba-based models.
>
> Furthermore, this structural interpretation is strongly supported by the variable-based ablation study presented in Table 3. When comparing several internal Mamba variables ($y_t$, $B_t$, $C_t$, and $\Delta_t$) under identical conditions, $\Delta_t$ consistently achieved the best and most stable pruning performance across all settings. This result empirically verifies that $\Delta_t$ is the most appropriate and reliable indicator for token importance.
>
> ---
> In the revised manuscript, we will emphasize this strengthened theoretical foundation, making it clear that $\Delta_t$ possesses sufficient validity and generality as a token importance metric. We hope that these additions will effectively address your concerns.

---

> ### Author Response · Authors · 2025-11-22
>
> ## Response to Weakness 3
>
> To begin with, we would like to emphasize that our work is focused on token pruning, and the scope of the paper is centered on identifying and removing visual tokens based on importance.
>
> Techniques such as token merging operate under a different objective, since they combine multiple tokens into a single representation rather than selecting tokens to prune based on their individual contribution. Because token merging focuses on combining tokens based on their representational similarity, it pursues a direction that is fundamentally different from importance-based pruning. Due to this difference, token merging does not align with the objective of our study.
>
> Regarding KV cache optimization, such approaches are specific to Transformer architectures, which rely on attention mechanisms and stored key value sequences.
> Mamba based MLLMs do not maintain key value caches and instead perform decoding using only the recurrent hidden state.
> As a result, Transformer oriented KV cache methods cannot be applied or meaningfully compared in this setting.
>
> As noted in lines 77–82 of the manuscript, most existing approaches are designed under the assumptions of Transformer architectures, and these structural dependencies make them difficult to apply directly to Mamba-based models. This architectural mismatch was, in fact, one of the motivations behind our work and led us to develop a new pruning strategy tailored specifically to the characteristics of Mamba.
>
> -----
> To broaden the baseline comparisons, we additionally include DART [1], a recent token reduction method that identifies redundant tokens using pivot selection and cosine similarity. Because DART does not rely on attention scores or key value caches, it can operate in both Transformer and Mamba based MLLMs.
>
> In our extended experiments on Cobra and RoboMamba, DTP consistently performs on par with or better than DART under matched FLOPs settings, reinforcing the effectiveness of the proposed approach. The experimental results are shown in the table below, and DTP consistently achieves the best performance, further clarifying the effectiveness of our proposed approach. We will add these results to Section 5.2 (MAIN RESULTS) of the revised manuscript.
>
>
> | Method                  | FLOPs | FLOPs Ratio | GQA  | VQA-v2 | TextVQA | POPE | VSR  | VizWiz | Avg  |
> |-------------------------|-------|-------------|------|--------|---------|------|------|--------|------|
> | Cobra                   | 2.01  | 100%        | 62.3 | 77.8   | 58.2    | 88.4 | 58.4 | 49.7   | 65.8 |
> |         |       |       |      |        |       |      |      |        |      |
> | FastV (k=2, r=0.7)      | 1.45  | 72%         | 62.1 | 77.4   | 56.9    | 87.7 | 58.0 | 49.8   | 65.3 |
> | VTW (K=45)              | 1.43  | 71%         | 62.1 | 77.7   | 58.2    | 88.3 | 58.5 | 49.5   | 65.7 |
> | DART (r=0.66)           | 1.38  | 68.7%       | 62.0 | 77.1   | 57.0    | 87.4 | 58.2 | 49.7   | 65.2 |
> | DTP (r=0.9)             | 1.35  | 67%         | 62.0 | 77.7   | 57.9    | 88.3 | 58.9 | 49.7   | 65.8 |
> |         |       |       |      |        |       |      |      |        |      |
> | FastV (k=2, r=0.5)      | 1.06  | 53%         | 61.7 | 76.8   | 55.0    | 87.4 | 57.3 | 50.1   | 64.7 |
> | VTW (K=32)              | 1.04  | 52%         | 47.1 | 54.1   | 42.6    | 74.1 | 57.9 | 48.5   | 54.0 |
> | DART (r=0.44)           | 0.96  | 48%         | 61.2 | 76.1   | 55.1    | 86.3 | 57.7 | 49.5   | 64.3 |
> | DTP (r=0.5)             | 0.97  | 48%         | 61.4 | 77.1   | 56.1    | 87.3 | 57.9 | 49.6   | 64.9 |
>
> | Method                  | FLOPs | FLOPs Ratio | OK-VQA | GQA  | VQA-v2 | POPE | VizWiz | Avg  |
> |-------------------------|-------|-------------|--------|------|--------|------|--------|------|
> | RoboMamba               | 0.70  | 100%        | 64.4   | 56.4 | 74.9   | 85.2 | 54.2   | 67.0 |
> |         |       |       |      |        |       |      |      |        |      |
> | FastV (k=2, r=0.7)      | 0.50  | 71%         | 64.1   | 56.1 | 74.4   | 85.3 | 53.0   | 66.6 |
> | VTW (K=45)              | 0.50  | 71%         | 64.0   | 55.9 | 74.7   | 85.3 | 53.3   | 66.6 |
> | DART (r=0.63)           | 0.47  | 67%         | 64.2   | 55.4 | 73.9   | 84.2 | 53.0   | 66.1 |
> | DTP (r=0.9)             | 0.46  | 66%         | 64.2   | 56.1 | 74.7   | 85.2 | 53.2   | 66.7 |
> |         |       |       |      |        |       |      |      |        |   |
> | FastV (k=2, r=0.5)      | 0.37  | 53%         | 63.4   | 55.1 | 73.4   | 84.2 | 52.1   | 65.6 |
> | VTW (K=32)              | 0.36  | 51%         | 40.0   | 44.7 | 55.2   | 82.1 | 45.2   | 53.4 |
> | DART (r=0.47)           | 0.36  | 51%         | 64.1   | 54.0 | 72.8   | 83.4 | 52.6   | 65.4 |
> | DTP (r=0.5)             | 0.34  | 49%         | 63.8   | 54.9 | 73.6   | 84.4 | 52.8   | 65.9 |
>
> [1] Wen, Zichen, et al. "Stop looking for important tokens in multimodal language models: Duplication matters more." arXiv preprint arXiv:2502.11494 (2025).

---

> ### Author Response · Authors · 2025-11-22
>
> ## Response to Weakness 2
> The cost of computing $\Delta_t$ is extremely small, taking approximately 0.01 ms in total. Importantly, $\Delta_t$ itself is already produced as part of Mamba’s computation, and the only additional operation required for DTP is a simple averaging over its dimensions.
>
> Even when incorporating both selective pruning and complete pruning, the overall overhead is roughly 0.1 ms, which is negligible compared to the total inference time.
>
> -----
> To evaluate efficiency beyond FLOPs, we conducted additional experiments on the Cobra model using the POPE to measure prefill latency, decode latency, total latency, and GPU memory usage.
>
> | Method            | FLOPs | Prefill Mean (ms) | Decode Mean (ms) | Total Latency | GPU Memory |
> |-------------------|-------|-------------------|-------------------------|----------------|------------|
> | Cobra (Original)  | 2.01  | 98.04             | 5.04                    | 16m 05s        | 8.8 GB     |
> | FastV (k=2, r=0.5)       | 1.06  | 60.64             | 4.94                | 10m 24s        | 8.5 GB     |
> | VTW (k=32)          | 1.04  | 67.31             | 4.99                    | 11m 25s        | 8.3 GB |
> | DART (r=0.44)        | 0.96  | 76.45             | 4.95                    | 12m 44s        | 8.5 GB     |
> | DTP (r = 0.5)     | 0.97  | 61.54             | 5.02                    | 10m 35s        | 8.3 GB     |
>
> The results show that our proposed method, DTP (r = 0.5), substantially reduces the prefill latency from 98.04 ms to 61.54 ms and decreases the total inference time from 16m 05s to 10m 35s, achieving an approximate 40% real-world speedup. GPU memory usage is also reduced compared to the original model. Notably, these improvements are competitive with, and in many cases superior to, existing token-pruning approaches such as FastV, VTW, and DART.
>
> Although decode latency does not change significantly, this behavior is expected and fully consistent with the computational characteristics of Mamba-based MLLMs, as described in Figure 1 and lines 50–51 of the original draft. Specifically, the prefill stage involves computation over the entire input sequence, whereas the decode stage generates each new token using only the previous hidden state and the most recently generated token. As a result, pruning has inherently limited influence on decode stage latency, and the decode time cost therefore remains essentially unchanged.
> Moreover, more than 95% of the total computational cost in Mamba-based MLLMs is incurred during the prefill stage, whereas the decode stage accounts for only a very small fraction of the overall latency. Consequently, even if decode latency were improved, its contribution to the total inference efficiency would be marginal compared to the dominant prefill cost.
>
> ----
> These inference latency results beyond FLOPs will be incorporated into the revised manuscript in Section 5.3 (Efficiency Analysis), clearly demonstrating that the FLOPs reduction achieved by DTP directly translates into substantial wall-clock performance improvement.

---

> ### Author Response · Authors · 2025-11-22
>
> ## Response to Question 1
>
> Yes. Extracting $\Delta_t$ in real inference pipelines is efficient and introduces virtually no additional latency.
>
> $\Delta_t$ is already computed as part of Mamba’s standard forward pass, and DTP only requires a lightweight averaging operation over its dimensions.
>
> The total cost of obtaining $\Delta_t$ is approximately 0.01 ms, and even with selective and complete pruning combined, the overall overhead remains around 0.1 ms, which is negligible relative to the total inference time.

---

> ### Author Response · Authors · 2025-11-22
>
> ## Response to Question 2
>
> We understand that the original draft may have appeared to select the 15th and 45th layers in a heuristic manner, which leads to questions about sensitivity and general applicability. In the revision, we will restructure Section 4.2 to present a general adaptive layer-selection procedure, where pruning layers are determined by analyzing the layer-wise standard deviation of $\Delta_t$-guided token importance rather than relying on fixed index values.
>
> -----
> The updated section now highlights the following generalizable procedure:
>
> **Selective pruning at early layer** is performed by locating the layer with the minimum standard deviation of $\Delta_t$-guided token importance within the first 25% of model depth. The validity of this criterion is further supported by the token importance distributions and implicit attention patterns, both of which consistently show clear clustering and strong token–token interactions near this region.
>
> $k_{\text{early}} = \arg\min_{\ell} Std_{\ell}$, where $0 \le \ell \le 0.25L$
>
> ----
> **Complete pruning at late layer** is performed by finding the layer that exhibits the largest inter-layer change in standard deviation within the final 30% of the depth. This point corresponds to the onset of the late layer cliff in the standard deviation curve. Beyond this region, implicit attention patterns show a marked weakening of token–token interactions, indicating that visual tokens no longer contribute stable or discriminative information to the computation. As a result, selective pruning becomes unreliable, and complete pruning becomes the more effective strategy.
>
> $k_{\text{late}} = \arg\max_{\ell} |Std_{\ell+1} - Std_{\ell}| + 1$, where $0.7L \le \ell < L-1$.
>
> ----
>
> Concretely, to apply this adaptive rule, we first draw a small subsample of the evaluation set for each model and compute the layer-wise standard deviation of the $\Delta_t$-guided token-importance distribution across this subset.
>
> We applied this adaptive rule across multiple benchmarks and both architectures, and the same layer-wise standard deviation patterns consistently appeared. When the adaptive rule is applied to Cobra and RoboMamba, it selects the 15th and 45th layers respectively. These index values are not manually chosen but instead emerge naturally from the underlying statistical patterns of the models.
>
> Since the optimal pruning layers consistently arise across datasets and architectures, DTP shows very low sensitivity to fixed pruning indices, and the adaptive rule provides a robust and generalizable mechanism for determining pruning layers.
>
> In the revised manuscript, we will clarify this adaptive procedure and include the corresponding layer-wise standard deviation plots in the appendix to make the robustness of layer selection more explicit.

---

> ### Author Response · Authors · 2025-11-22
>
> ## Response to Weakness 4
>
> Thank you for this helpful suggestion. We agree that the interpretation of some figures and analysis can be further strengthened to more clearly communicate their practical relevance.
>
> -----
> We clarified and refined the adaptive layer-selection procedure based on the layer-wise standard deviation of $\Delta_t$-guided token importance, and this revision was introduced in our response to Question 2.
> Building on this improved structural explanation, the revised manuscript will also enhance the interpretation of the implicit attention analysis to more clearly connect these quantitative patterns with the qualitative visualizations.
>
> -----
> In the revised manuscript, we will enhance the explanation of the implicit attention analysis by explicitly describing the correspondence between the visual patterns of implicit attention and the quantitative $\Delta_t$-guided statistics. We will also revise the associated figures by adding clearer annotations that highlight the local minimum peak in the early-layer region and the cliff in the late layers where the standard deviation rises abruptly.
>
> Across both Cobra and RoboMamba, we observe that layers exhibiting low standard deviation in $\Delta_t$-guided token importance consistently show strong token–token interaction patterns, whereas layers entering the high standard deviation region display weak interactions. This alignment between the qualitative implicit attention maps and the quantitative layer-wise standard deviation helps clarify why these regions serve as the basis for selective and complete pruning.
>
> -----
> We hope that these revisions will make the analysis easier to understand and help readers more clearly grasp its practical implications.
>
> We will update the revised manuscript as promptly as possible.

---

> ### Author Response · Authors · 2025-11-22
>
> ## Response to Question 3
>
> We appreciate your suggestion and the idea of exploring whether $\Delta_t$-based importance could be integrated with gradient-based saliency.
>
> This is a meaningful direction, and to better understand its potential, we examined the recent gradient-driven token pruning method SDTP [1], which provides a representative example of how gradient attribution can be leveraged for estimating token importance.
> SDTP computes token saliency via backward propagation and then trains a lightweight pruning module using MSE and ranking losses to dynamically retain informative tokens during inference. This approach demonstrates strong performance within a supervised training setup.
>
> In our work, however, we focus on a **training-free** pruning strategy guided by $\Delta_t$-based token importance. Because our goal is to design a fully inference-time method without relying on gradient signals or additional learned modules, incorporating gradient-based supervision would require a training process and shift the method away from this objective. In contrast, the $\Delta_t$-based score arises naturally from the model’s internal behavior and already provides a stable and lightweight signal for identifying salient tokens without requiring gradient computation.
>
> -----
> In summary, although combining $\Delta_t$-based importance with gradient-based saliency is an interesting direction, such integration does not naturally align with the training-free objective of our approach.
>
> Exploring hybrid strategies may be worthwhile in future work, but it would require additional design considerations beyond the scope of the present study.
>
> -----
> [1] Tao, Y., Tang, Y., Wang, Y., Zhu, M., Hu, H., & Wang, Y. (2025). Saliency-driven dynamic token pruning for large language models. arXiv preprint arXiv:2504.04514.

---

> ### Author Response · Authors · 2025-11-22
>
> ## Response to Question 4
> Thank you for the question. To thoroughly assess the behavior of DTP under extremely high pruning settings, we expanded our evaluation to include pruning ratios ranging from r = 1.0 down to r = 0.3, which corresponds to retaining as little as 30% of the visual tokens.
>
> This extension builds upon the original experiments which used r = 0.9 and r = 0.5 and now provides a broader and more systematic characterization of DTP across different pruning ratio regimes.
> The results for both Cobra and RoboMamba are shown below.
>
> | Method            | FLOPs | FLOPs Ratio | GQA   | VQA-v2 | TextVQA | POPE  | VSR   | VizWiz | Avg   |
> |-------------------|-------|-------------|-------|--------|---------|-------|-------|--------|-------|
> | Cobra (r = 1.0)   | 2.01  | 100%        | 62.3  | 77.8   | 58.2    | 88.4  | 58.4  | 49.7   | 65.8  |
> | Cobra (r = 0.9)   | 1.35  | 67%         | 62.0  | 77.7   | 57.9    | 88.3  | 58.9  | 49.7   | 65.8  |
> | Cobra (r = 0.7)   | 1.16  | 58%         | 62.2 | 77.6   | 57.4    | 87.9  | 58.3  | 49.9   | 65.5  |
> | Cobra (r = 0.5)   | 0.97  | 48%         | 61.4  | 77.1   | 56.1    | 87.3  | 57.9  | 49.6   | 64.9  |
> | Cobra (r = 0.3)   | 0.79  | 39%         | 60.0  | 75.7   | 53.2    | 86.3  | 57.9  | 50.0   | 63.8  |
>
> | Method                | FLOPs | FLOPs Ratio | OK-VQA | GQA   | VQA-v2 | POPE  | VizWiz | Avg   |
> |-----------------------|-------|-------------|--------|-------|--------|-------|--------|-------|
> | RoboMamba (r = 1.0)   | 0.70  | 100%        | 64.4   | 56.4  | 74.9   | 85.1  | 54.2   | 67.0  |
> | RoboMamba (r = 0.9)   | 0.46  | 66%         | 64.2   | 56.1  | 74.7   | 85.2  | 53.2   | 66.7  |
> | RoboMamba (r = 0.7)   | 0.40  | 57%         | 64.4   | 56.0  | 74.4   | 85.0  | 53.5   | 66.7  |
> | RoboMamba (r = 0.5)   | 0.34  | 49%         | 63.8   | 54.9  | 73.6   | 84.4  | 52.8   | 65.9  |
> | RoboMamba (r = 0.3)   | 0.27  | 39%         | 63.4   | 52.9  | 71.7  | 82.8 | 51.7   | 64.5  |
>
> ----
> Across all pruning strengths including the extremely high pruning ratios r = 0.3 configuration performance degrades smoothly and monotonically without extrem collapse.
>
> Notably, even at r = 0.3, both models maintain around 96% of their original accuracy, while operating at only 39% of the baseline FLOPs. This demonstrates that DTP remains stable and effective even when a significant majority of visual tokens are removed.
>
> The full results will be included in the appendix of the revised manuscript.

---

> ### Author Response · Authors · 2025-11-22
>
> ## Response to Question 5
>
> Yes. After noticing that the implicit attention visualization for RoboMamba was missing from the appendix, we will add the corresponding figures in the revised version.
>
> The updated results confirm that both Cobra and RoboMamba exhibit highly consistent implicit attention behaviors. In particular, the regions where the layer-wise standard deviation of $\Delta_t$-guided token importance is low consistently show strong token–token interactions, whereas regions with high standard deviation show noticeably weaker interactions. This agreement across architectures indicates that Mamba-based MLLMs learn similar layer-dependent interaction patterns during training.
>
> We will incorporate these visualizations and clarifications into the revised appendix to clearly document the observed consistency.
>
> -----
> Your comments have guided us toward several important improvements, and we are in the process of reflecting these enhancements in the revised manuscript. We will update the revised manuscript as promptly as possible.
>
> We hope that the clarifications and additional results provided in this response help resolve the concerns you raised, and we would be sincerely grateful if you could consider these revisions when re-evaluating our work.
>
> If there are any further points you would like to discuss or additional suggestions you wish to offer, please feel free to let us know at any time. Your feedback is always welcome, and we remain committed to continuously improving this research based on your insightful comments.

---

### Official Review · Reviewer_K7EU · 2025-10-31

**Soundness:** 2
**Presentation:** 3
**Contribution:** 2
**Rating:** 4
**Confidence:** 3

**Summary:**

This paper introduces Delta-guided Two-stage Pruning (DTP), a training-free framework for visual token pruning in Mamba-based multimodal large language models.

**Strengths:**

[1] Adaptation of pruning for Mamba’s state-space mechanism, clearly distinct from Transformer-specific attention-based approaches.
[2] Two-stage pruning strategy is empirically well-motivated by variance and implicit-attention statistics.
[3] Extensive ablations validating design choices and demonstrating robustness.
[4] The method performs pruning during inference without any retraining or fine-tuning, which makes it practically deployable.

**Weaknesses:**

[1] The layer selection heuristic (15th & 45th) is empirical; a more formal justification or adaptive strategy could strengthen generality.
[2] The analysis depth of implicit attention remains qualitative; quantitative correlation with pruning behavior would improve rigor.
[3] While FLOPs reduction is well-documented, real-world inference latency (wall-clock time) is not reported.
[4] Limited baselines are included. There are many works that should be considered for comparison, e.g. PyramidKV [a], VL-cache [b], etc.
[5] Novelty is limited, where the methods/motivation are borrowed from the transformer-based research work.


[a] Cai, Zefan, et al. "Pyramidkv: Dynamic kv cache compression based on pyramidal information funneling." arXiv preprint arXiv:2406.02069 (2024).
[b]   Tu, Dezhan, et al. "VL-cache: Sparsity and modality-aware KV cache compression for vision-language model inference acceleration." arXiv preprint arXiv:2410.23317 (2024).

**Questions:**

How were the 15th and 45th layers chosen for selective and complete pruning? Were these empirically optimal for all models or do they depend on model depth or dataset characteristics?

---

> ### Author Response · Authors · 2025-11-21
>
> We sincerely appreciate the reviewer’s constructive feedback and critical perspective. In response, we provide the following clarification addressing the concern.
>
> ___
> ## Response to Weakness 1
> We sincerely appreciate the reviewer’s insightful comment regarding the empirical nature of the 15th and 45th layer selection. We fully agree that relying solely on fixed layer indices may limit generality, and that providing a more principled and adaptive formulation can strengthen the applicability of our method.
>
> To address this, we will revise the manuscript to clarify that the layers reported for Cobra and RoboMamba are not fixed heuristics, but concrete instances of a fully general, model adaptive two-stage pruning strategy.
>
> ----
> The revision will present the following generalizable procedure:
>
> **Selective pruning at early layer** is performed by locating the layer with the minimum standard deviation of $\Delta_t$-guided token importance within the first 25% of model depth. The validity of this criterion is further supported by the token importance distributions and implicit attention patterns, both of which consistently show clear clustering and strong token–token interactions near this region.
>
> $k_{\text{early}} = \arg\min_{\ell} Std_{\ell}$, where $0 \le \ell \le 0.25L$
>
> **Complete pruning at late layer**is performed by finding the layer that exhibits the largest inter-layer change in standard deviation within the final 30% of the depth. This point corresponds to the onset of the late layer cliff in the standard deviation curve. Beyond this region, implicit attention patterns show a marked weakening of token–token interactions, indicating that visual tokens no longer contribute stable or discriminative information to the computation. As a result, selective pruning becomes unreliable, and complete pruning becomes the more effective strategy.
>
> $k_{\text{late}} = \arg\max_{\ell} |Std_{\ell+1} - Std_{\ell}| + 1$, where $0.7L \le \ell < L-1$.
>
> ----
> Across all benchmark datasets and for both Cobra and RoboMamba, we consistently observed the same patterns of layer-wise standard deviation, providing strong empirical support that these criteria generalize well beyond the specific models studied. We will further supplement the manuscript by adding the layer-wise standard deviation plots in the appendix, ensuring that these observations are transparently documented.
> We believe this formal justification directly addresses your concern and improves the clarity and generality of our method.

---

> ### Author Response · Authors · 2025-11-21
>
> ## Response to Weakness 2
> Thank you for this valuable comment. Although implicit attention is qualitative, in our framework it is intentionally used as a complementary visualization that is closely connected to a clear quantitative signal, namely the layer-wise standard deviation of delta-guided token importance. The implicit attention analysis is not meant to function independently. Instead, it reinforces and visually contextualizes the quantitative patterns that determine the pruning layers.
>
> The attention maps consistently reflect the same depth-dependent trends revealed by the standard deviation curve. Token to token interactions appear stronger in the early layer region where the standard deviation reaches a local low standard deviation peak, and they become substantially weaker in the late layer region after the standard deviation increases sharply. These qualitative patterns align with the quantitative layer-wise statistics and serve as an intuitive visual counterpart to the pruning signals.
>
> -----
> To improve clarity and rigor, we will revise the manuscript to explain this relationship more explicitly and clarify that the qualitative implicit attention patterns do not simply accompany the quantitative analysis but meaningfully complement it by validating and intuitively reinforcing the statistical criteria used to determine pruning layers. We believe that this strengthened explanation will enhance both the rigor and interpretability of the analysis and fully address the reviewer’s concern regarding the need for clearer quantitative correlation.

---

> ### Author Response · Authors · 2025-11-21
>
> ## Response to Weakness 4
> Thank you for raising this important point. The methods you suggested, PyramidKV and VL-Cache, are fundamentally designed for Transformer-based architectures and rely on manipulating the KV cache. Since Mamba does not use any KV cache by design, these approaches are not directly applicable to Mamba-based Multimodal Large Language Models.
>
> As noted in lines 77–82 of the manuscript, most existing approaches are designed under the assumptions of Transformer architectures, and these structural dependencies make them difficult to apply directly to Mamba-based models.
> This architectural mismatch was, in fact, one of the motivations behind our work and led us to develop a new pruning strategy tailored specifically to the characteristics of Mamba.
> Our method is based on a detailed analysis of internal parameters of mamba, as well as layer-wise statistical examination of token importance distributions and implicit attention patterns, allowing us to identify layers that are most suitable for pruning. This leads to a fundamentally different approach from Transformer-based pruning methods.
>
> ___
> To further reflect the reviewer’s suggestion, we additionally include DART (Duplication-Aware Reduction of Tokens) [1], one of the recent token reduction methods, as an additional baseline. Since DART detects redundant tokens by selecting pivots and measuring cosine similarity, it can be applied in architectures beyond Transformers, including Mamba-based models. As shown in the table below, our DTP method consistently outperforms DART across benchmarks, further supporting the effectiveness of the proposed approach.
>
> We will add these results to Section 5.2 (MAIN RESULTS) of the revised manuscript.
> | Method                  | FLOPs | FLOPs Ratio | GQA  | VQA-v2 | TextVQA | POPE | VSR  | VizWiz | Avg  |
> |-------------------------|-------|-------------|------|--------|---------|------|------|--------|------|
> | Cobra                   | 2.01  | 100%        | 62.3 | 77.8   | 58.2    | 88.4 | 58.4 | 49.7   | 65.8 |
> |         |       |             |      |        |         |      |      |        |      |
> | FastV (k=2, r=0.7)      | 1.45  | 72%         | 62.1 | 77.4   | 56.9    | 87.7 | 58.0 | 49.8   | 65.3 |
> | VTW (K=45)              | 1.43  | 71%         | 62.1 | 77.7   | 58.2    | 88.3 | 58.5 | 49.5   | 65.7 |
> | DART (r=0.66)           | 1.38  | 68.7%       | 62.0 | 77.1   | 57.0    | 87.4 | 58.2 | 49.7   | 65.2 |
> | DTP (r=0.9)             | 1.35  | 67%         | 62.0 | 77.7   | 57.9    | 88.3 | 58.9 | 49.7   | 65.8 |
> |         |       |             |      |        |         |      |      |        |      |
> | FastV (k=2, r=0.5)      | 1.06  | 53%         | 61.7 | 76.8   | 55.0    | 87.4 | 57.3 | 50.1   | 64.7 |
> | VTW (K=32)              | 1.04  | 52%         | 47.1 | 54.1   | 42.6    | 74.1 | 57.9 | 48.5   | 54.0 |
> | DART (r=0.44)           | 0.96  | 48%         | 61.2 | 76.1   | 55.1    | 86.3 | 57.7 | 49.5   | 64.3 |
> | DTP (r=0.5)             | 0.97  | 48%         | 61.4 | 77.1   | 56.1    | 87.3 | 57.9 | 49.6   | 64.9 |
>
> | Method                  | FLOPs | FLOPs Ratio | OK-VQA | GQA  | VQA-v2 | POPE | VizWiz | Avg  |
> |-------------------------|-------|-------------|--------|------|--------|------|--------|------|
> | RoboMamba               | 0.70  | 100%        | 64.4   | 56.4 | 74.9   | 85.2 | 54.2   | 67.0 |
> |         |       |             |      |        |         |      |      |        |      |
> | FastV (k=2, r=0.7)      | 0.50  | 71%         | 64.1   | 56.1 | 74.4   | 85.3 | 53.0   | 66.6 |
> | VTW (K=45)              | 0.50  | 71%         | 64.0   | 55.9 | 74.7   | 85.3 | 53.3   | 66.6 |
> | DART (r=0.63)           | 0.47  | 67%         | 64.2   | 55.4 | 73.9   | 84.2 | 53.0   | 66.1 |
> | DTP (r=0.9)             | 0.46  | 66%         | 64.2   | 56.1 | 74.7   | 85.2 | 53.2   | 66.7 |
> |         |       |             |      |        |         |      |      |        |      |
> | FastV (k=2, r=0.5)      | 0.37  | 53%         | 63.4   | 55.1 | 73.4   | 84.2 | 52.1   | 65.6 |
> | VTW (K=32)              | 0.36  | 51%         | 40.0   | 44.7 | 55.2   | 82.1 | 45.2   | 53.4 |
> | DART (r=0.47)           | 0.36  | 51%         | 64.1   | 54.0 | 72.8   | 83.4 | 52.6   | 65.4 |
> | DTP (r=0.5)             | 0.34  | 49%         | 63.8   | 54.9 | 73.6   | 84.4 | 52.8   | 65.9 |
>
> [1] Wen, Zichen, et al. "Stop looking for important tokens in multimodal language models: Duplication matters more." arXiv preprint arXiv:2502.11494 (2025).

---

> ### Author Response · Authors · 2025-11-21
>
> ## Response to Weakness 3
> Thank you for pointing out that reporting only FLOPs, without real-world inference latency.
> In response, we conducted additional experiments on the Cobra model using the POPE to measure real-world inference latency including prefill latency, decode latency, total latency, and GPU memory usage.
>
> | Method            | FLOPs | Prefill Mean (ms) | Decode Mean (ms) | Total Latency | GPU Memory |
> |-------------------|-------|-------------------|-------------------------|----------------|------------|
> | Cobra (Original)  | 2.01  | 98.04             | 5.04                    | 16m 05s        | 8.8 GB     |
> | FastV (k=2, r=0.5)       | 1.06  | 60.64             | 4.94                | 10m 24s        | 8.5 GB     |
> | VTW (k=32)          | 1.04  | 67.31             | 4.99                    | 11m 25s        | 8.3 GB |
> | DART (r=0.44)        | 0.96  | 76.45             | 4.95                    | 12m 44s        | 8.5 GB     |
> | DTP (r = 0.5)     | 0.97  | 61.54             | 5.02                    | 10m 35s        | 8.3 GB     |
>
> The results show that our proposed method, DTP (r = 0.5), substantially reduces the prefill latency from 98.04 ms to 61.54 ms and decreases the total inference time from 16m 05s to 10m 35s, achieving an approximate 40% real-world speedup. GPU memory usage is also reduced compared to the original model. Notably, these improvements are competitive with, and in many cases superior to, existing token-pruning approaches such as FastV, VTW, and DART.
>
> Although decode latency does not change significantly, this behavior is expected and fully consistent with the computational characteristics of Mamba-based MLLMs, as described in Figure 1 and lines 50–51 of the original draft. Specifically, the prefill stage involves computation over the entire input sequence, whereas the decode stage generates each new token using only the previous hidden state and the most recently generated token. As a result, pruning has inherently limited influence on decode stage latency, and the decode time cost therefore remains essentially unchanged.
> Moreover, more than 95% of the total computational cost in Mamba-based MLLMs is incurred during the prefill stage, whereas the decode stage accounts for only a very small fraction of the overall latency. Consequently, even if decode latency were improved, its contribution to the total inference efficiency would be marginal compared to the dominant prefill cost.
>
> ----
> These real-world inference latency results will be incorporated into the revised manuscript in Section 5.3 (Efficiency Analysis), clearly demonstrating that the FLOPs reduction achieved by DTP directly translates into substantial wall-clock performance improvement.

---

> ### Author Response · Authors · 2025-11-21
>
> ## Response to Weakness 5
> Thank you for raising this point.
>
> The central motivation of our work stems from the fact that principles used in transformer-based pruning cannot be applied to Mamba due to the fundamentally different computational structure of the two architectures. Transformer pruning techniques rely on attention scores, key–value cache manipulations, or attention-driven token importance, none of which have functional directly counterparts in Mamba.
> Because of this architectural discrepancy, transformer-based methods cannot be directly borrowed for Mamba, as their attention-based mechanisms do not directly translate into usable criteria for layer selection, token importance evaluation, or pruning behavior in state-space models.
>
> Our method is not a borrowing of transformer techniques, but a new formulation developed specifically for Mamba. We design a structured pruning method tailored to Mamba, incorporating delta-guided token importance, layer-wise statistical analysis, and layer-adaptive pruning criteria.
>
> To the best of our knowledge, this is the first token pruning method developed for Mamba-based multimodal large language models, and this novelty directly corresponds to the reviewer’s observation that our approach is clearly distinct from transformer-specific attention-driven methods.
>
> -----
> In the revised manuscript, we will further highlight this distinction and clarify that the contribution of our work lies in establishing a pruning methodology for an architectural regime where existing transformer-based techniques simply do not apply.
> We believe this clarification meaningfully addresses the reviewer’s concern regarding novelty.

---

> ### Author Response · Authors · 2025-11-21
>
> ## Response to Question 1
> Thank you for your valuable question.
>
> In the previous version of the manuscript, the 15th and 45th layers were presented in a way that could appear heuristic. To improve formal justification and strengthen generality, the revised manuscript now derives these pruning layers through an adaptive, statistically grounded procedure rather than fixed indices.
>
> Specifically, the pruning positions are determined by analyzing the layer-wise standard deviation of delta-guided token importance.
> For selective pruning at the early stage, we identify the layer within the first 25% of the depth where the standard deviation reaches its minimum. This depth consistently marks the point at which meaningful separation between informative and redundant tokens first emerges. For complete pruning at the late stage, we select the layer within the final 30% of depth where the inter-layer change in standard deviation is maximal, corresponding to the onset of the variance cliff where token–token interactions break down and visual tokens no longer provide stable information.
>
> Applying this adaptive procedure to Cobra and RoboMamba leads to the 15th and 45th layers observed in our experiments. These layers emerge naturally from the formalized selection rule rather than from any predefined heuristic.
>
> Importantly, across all evaluation datasets and for both Cobra and RoboMamba, we consistently observed the same depth-normalized patterns in the layer-wise standard deviation curves. This convergence strongly supports that the revised formulation provides a reliable, principled, and broadly generalizable strategy for determining pruning layers in Mamba-based models.
>
> -----
> Your comments have guided us toward several important improvements, and we are in the process of reflecting these enhancements in the revised manuscript.
>
> We hope that the clarifications and additional results provided in this response help resolve the concerns you raised, and we would be sincerely grateful if you could consider these revisions when re-evaluating our work.
>
> If there are any further points you would like to discuss or additional suggestions you wish to offer, please feel free to let us know at any time. Your feedback is always welcome, and we remain committed to continuously improving this research based on your insightful comments.

---

### Author Response · Authors · 2025-11-29
**Summary of Revisions and Responses**

**Dear Reviewers and Area Chair,**

Thank you very much for the insightful and constructive feedback. The manuscript has been updated to address the concerns raised by the reviewers, and several improvements have been incorporated accordingly.

The main updates are as follows:

- Introducing a more generalizable formulation for the proposed pruning layer selection strategy.

- Adding new token-pruning baselines to strengthen comparative evaluation.

- Reporting real-world efficiency metrics, including latency and GPU memory usage, in addition to FLOPs.

- Refining figures and analyses to enhance clarity and interpretability.

The thoughtful comments and suggestions have been highly valuable in improving the quality and clarity of the work, and we deeply appreciate the reviewers’ efforts and time.

---

> ### Author Response · Authors · 2025-11-29
>
> Below, we provide a consolidated summary of the strengths and key concerns commonly raised across multiple reviewers, along with our corresponding responses.The summary highlights overlapping points rather than listing every individual comment.
> We have also provided detailed responses to all reviewer-specific weaknesses and questions, and these can be found in the individual replies to each reviewer.
>
> -----
> ## **Strengths**
>
>  **1. A pruning paradigm tailored to Mamba** (Reviewers K7EU, BGBy and oGU1)
>
> - A novel pruning approach aligned with Mamba’s state-space mechanism and clearly distinct from Transformer-based methods.
>
>  **2. Well-motivated two-stage pruning strategy** (Reviewers K7EU, BGBy and pdx2)
>
> - The early selective pruning and late complete pruning design is intuitive and strongly validated by empirical evidence.
> -----
> ## **Weaknesses and Responses**
>
> **1. Limited generality and justification of pruning layer selection** (Reviewers K7EU, oGU1, and pdx2)
>   - The reviewers pointed out that selecting the 15th and 45th layers empirically may limit generality across Mamba models with different depths. To address this, we clarified in the revised manuscript that these layers are not fixed heuristics but empirical instances of a fully general, model-adaptive layer selection recipe.
>   - The generalized layer selection procedure is summarized below:
>
>     - Selective pruning at early layer:
>
>       - The early pruning layer is selected as the layer with the minimum standard deviation of $\Delta_t$-guided token importance within the first 25% of model depth. This region consistently shows strong token–token interactions.
>
>           $k_{\text{early}} = \arg\min_{\ell} Std_{\ell}$, where $0 \le \ell \le 0.25L$
>
>
>     - Complete pruning at late layer:
>
>       - The late pruning layer is determined by the point with the largest inter-layer change in standard deviation within the final 30% of the depth. This region consistently shows weak token–token interactions.
>
>           $k_{\text{late}} = \arg\max_{\ell} |Std_{\ell+1} - Std_{\ell}| + 1$, where $0.7L \le \ell < L-1$.
>
>   - Consistent standard deviation patterns across all datasets and across both Cobra and RoboMamba strongly support the generality of this approach. Layer-wise standard deviation plots added to the appendix for transparency and reproducibility.
>   - These clarifications and additions are reflected in the revised manuscript to improve the formal rigor and general applicability of our pruning strategy.
>
>
> **2. Incomplete evaluation of real-world efficiency beyond FLOPs** (Reviewers K7EU, BGBy, oGU1, and pdx2)
>
>    - We conducted additional experiments on the Cobra model using the POPE benchmark to measure prefill latency, decode latency, total inference time, and GPU memory usage.
>
>    - Key results include:
>
>       - Prefill latency: 98.04 ms → 61.54 ms
>
>      - Total inference time: 16m 05s → 10m 35s
>
>      - GPU memory: 8.8 GB → 8.3 GB
>
>
>    - These measurements directly demonstrate that the FLOPs reduction achieved by DTP translates into substantial wall-clock gains.
>    We include these results in Section 5.3 (Efficiency Analysis) of the revised manuscript.
>
> -----

---

### Meta-Review · Area_Chair_JhZC · 2026-01-06

**Summary:**

Reviewers raised concerns about the heuristic choice of pruned layers (e.g., fixed selection of the 15th/45th layers), lack of real-world latency measurements, insufficient comparison with recent baselines, and limited justification for the method’s applicability beyond Mamba architectures. They also questioned the statistical robustness of token importance estimation and the generality of the proposed two-stage pruning framework.

**Reviewer Concerns:**

The rebuttal adequately addressed concerns regarding heuristic layer selection by introducing a statistically grounded criterion based on token importance variance, and added real-world latency results to support efficiency claims. It also incorporated additional baselines and clarified the method’s scope. However, lingering concerns remain about the generalizability of DTP beyond Mamba architectures and whether the two-stage design truly captures dynamic token redundancy across diverse tasks. Some reviewers may still question the robustness of token importance estimation under varying data distributions. Overall, I appreciate the authors' rebuttal and believe that most of the concerns are addressed.

**Reviewer Scores:**

One reviewer states that the score will be improved, maybe from 4 to 6. While other 3 reviewers didn't participate the discussion.

---

### Decision · Program_Chairs · 2026-01-26

Accept (Poster)